# Inter-comparison study of atmospheric $^{222}$Rn and $^{222}$Rn progeny monitors

Claudia Grossi[1,2], Scott D. Chambers[3], Olivier Llido[4], Felix R. Vogel[5], Victor Kazan[4], Alessandro Capuana[6], Sylvester Werczynski[3], Roger Curcoll[7,8], Marc Delmotte[4], Arturo Vargas[1], Josep-Anton Morguí[7,9], Ingeborg Levin[6], Michel Ramonet[4].

[1] Institut de Tècniques Energètiques (INTE), Universitat Politècnica de Catalunya (UPC), Barcelona, Spain;

[2] Physics Department, Universitat Politècnica de Catalunya (UPC), Barcelona, Spain;

[3] Environmental Research, ANSTO, Lucas Heights, Australia;

[4] Laboratoire des Sciences du Climat et de l'Environnement, Université Paris-Saclay (LSCE/IPSL, CEA-CNRS-UVSQ), Gif-sur-Yvette, France;

[5] Climate Research Division, Environment and Climate Change Canada, Toronto, Canada;

[6] Institut für Umweltphysik (IUP), Heidelberg University, Heidelberg, Germany;

[7] Institut de Ciència i Tecnologia Ambientals (ICTA), Universitat Autònoma de Barcelona (UAB), Cerdanyola del Vallès, Spain;

[8] Chemical Department, Universitat Politècnica de Catalunya (UPC), Barcelona, Spain;

[9] Departament Biologia Evolutiva, Ecologia i Ciències Ambientals, Universitat de Barcelona (UB), Barcelona, Spain.

*Correspondence to*: Claudia Grossi (claudia.grossi@upc.edu)

**Abstract.**

The use of the noble gas radon ($^{222}$Rn) as tracer for different research studies, for example observation-based estimation of greenhouse gas (GHG) fluxes, has led to the need of high-quality $^{222}$Rn activity concentration observations with high spatial and temporal resolution. So far a robust metrology chain for these measurements is not yet available.

A portable direct Atmospheric Radon MONitor (ARMON), based on electrostatic collection of $^{218}$Po, is nowadays running at Spanish stations. This monitor has not yet been compared with other $^{222}$Rn and $^{222}$Rn progeny monitors commonly used at atmospheric stations.

A 3-month inter-comparison campaign of atmospheric $^{222}$Rn and $^{222}$Rn progeny monitors based on different measurement techniques was realized during the fall and winter of 2016-2017 to evaluate: i) calibration and correction factors between monitors necessary to harmonize the atmospheric radon

observations; and ii) the dependence of each monitor's response in relation to the sampling height,
meteorological and atmospheric aerosol conditions.
Results of this study have shown that: i) all monitors were able to reproduce the atmospheric radon
variability on daily basis; ii) linear regression fits between the monitors exhibited slopes, representing the
correction factors, between 0.62 and 1.17 and offsets ranging between -0.85 Bq m$^{-3}$ and -0.23 Bq m$^{-3}$
when sampling 2 m above ground level (a.g.l.). Corresponding results at 100 m a.g.l. exhibited slopes of
0.94 and 1.03 with offsets of -0.13 Bq m$^{-3}$ and 0.01 Bq m$^{-3}$, respectively; iii) no influence of atmospheric
temperature and relative humidity on monitor responses was observed for unsaturated conditions at 100 m
a.g.l. whereas slight influences (order of $10^{-2}$) of ambient temperature were observed at 2 m a.g.l.; iv)
changes of the ratio between $^{222}$Rn progeny and $^{222}$Rn monitor responses were observed under very low
atmospheric aerosol concentrations.
Results also show that the new ARMON could be useful at atmospheric radon monitoring stations with
space restrictions or as a mobile reference instrument to calibrate in situ $^{222}$Rn progeny monitors and fixed
radon monitors. In the near future a long-term comparison study between ARMON, HRM and ANSTO
monitors would be useful to better evaluate: i) the uncertainties of radon measurements in the range of a
few hundred mBq m$^{-3}$ to a few Bq m$^{-3}$; and ii) the response time correction of the ANSTO monitor for
representing fast changes in the ambient radon concentrations.
Key words: radon, activity concentration, atmosphere, one-filter, two-filters, electrodeposition
**1 Introduction**
Over continents, the natural radioactive noble gas radon ($^{222}$Rn) (half-life T$_{1/2}$ = 3.8 days) is continuously
generated within the soil from the decay of radium ($^{226}$Ra) (Nazaroff and Nero, 1988; Porstendörfer,
1994) and it can then escape into the atmosphere by diffusion, depending on soil characteristics and
meteorological conditions (Grossi et al., 2011, Lopez-Coto et al., 2013; Karstens et al., 2015). The global
$^{222}$Rn source into the atmosphere is mainly restricted to land surfaces (Szegvary et al., 2009; Karstens et
al., 2015), with the $^{222}$Rn flux from water surfaces considered negligible for most applications (Schery
and Huang, 2004).
In recent decades the atmospheric scientific community has been addressing different research topics
using $^{222}$Rn as a tracer. Examples of such applications include: the improvement of inverse transport
models (Hirao et al., 2010), the improvement of chemical transport models (Jacob and Prather, 1990;
Chambers et al. 2019a), the study of atmospheric transport and mixing processes within the planetary
boundary layer (Zahorowski et al., 2004; Galmarini, 2006; Baskaran, 2011; Chambers et al., 2011, 2019b;
Williams et al., 2011, 2013; Vogel et al. 2013; Vargas et al., 2015; Baskaran, 2016), the experimental
estimation of greenhouse gas (GHG) fluxes (Levin et al., 1999; 2011; Vogel et al., 2012; Wada et al.,
2013; Grossi et al., 2018), and others listed in Grossi et al. (2016).
In light of this, atmospheric $^{222}$Rn measurements are being carried out at numerous monitoring stations of
GHG concentrations and air quality using three fundamentally different measurement principles: one
filter, two filters, and electrostatic deposition (Stockburger and Sittkus, 1966; Polian, 1986; Hopke, 1989;
Whittlestone and Zahorowski, 1998; Paatero et al., 1998; Levin et al., 2002). The two most commonly
employed measurement systems at European $^{222}$Rn monitoring stations are: the dual-flow-loop two-filter
monitor (Whittlestone and Zahorowski, 1998; Zahorowski et al. 2004; Chambers et al., 2011, 2014,
2018; Griffith et al., 2016), which samples and measures radon directly, and the one-filter monitors, of
which several kinds are in use (e.g. Stockburger and Sittkus, 1966; Polian, 1986; Paatero et al., 1998;
Levin et al., 2002), which sample and measure aerosol-bound radon progeny. Finally, a third method is
being used at several Spanish atmospheric stations (Vargas et al., 2015; Hernández-Ceballos et al., 2015;
Grossi et al., 2016; Frank et al., 2016; Grossi et al., 2018; Gutiérrez-Álvarez et al., 2019). This type of
instrument performs a direct measurement of $^{222}$Rn and $^{220}$Rn (thoron) activity concentrations using the
already existent method based on the electrostatic deposition of $^{218}$Po and $^{216}$Po, respectively (Hopke,
1989; Tositti et al., 2002; Grossi et al., 2012).
The diversity of these three aforementioned measurement techniques could introduce biases or
compatibility issues that would limit the comparability of the results obtained by independent studies and
the subsequent application of atmospheric radon data for regional-to-global investigations (e.g.
Schmithüsen et al., 2017). Thus, a comparative assessment of all the experimental techniques applied for
atmospheric $^{222}$Rn activity concentration measurements and a harmonization of their datasets is needed, as
suggested by the International Atomic Energy Agency (IAEA, 2012).
Xia et al. (2010) carried out a comparison of the response of a dual-flow-loop two-filter detector from the
Australian Nuclear Science and Technology Organisation (ANSTO, Whittlestone and Zahorowski 1998)
and a one-filter monitor (α/β Monitor P3) manufactured by the Bundesamt für Strahlenschutz, Germany
(BfS) (Stockburger and Sittkus, 1966), for atmospheric $^{222}$Rn measurements under various meteorological
conditions at 2.5 m above ground level (a.g.l.) over one year. Their results showed that both systems
followed the same patterns and produced very similar results most of the time, except under specific
meteorological conditions such as when precipitation or the proximity of the forest canopy could remove
short-lived progeny from the air mass to be measured by the one-filter monitor. However, Xia et al.
(2010) did not find a clear relationship between precipitation intensity and the ratio between progeny-
derived $^{222}$Rn and $^{222}$Rn activity concentration to convert the progeny signal to $^{222}$Rn activity
concentration.
Grossi et al. (2016) presented results from two short (about 7-9 days) comparisons between a one-filter
monitor from Heidelberg University (HRM; Levin et al., 2002), and an Atmospheric Radon MONitor
(ARMON, Grossi et al., 2012), an electrostatic deposition monitor from the Universitat Politecnica de
Catalunya (UPC). The two comparison campaigns were carried out at a coastal and a mountain site, with
sampling in both cases from 10 m a.g.l. These comparisons revealed that the responses of both monitors
were in agreement except for water saturated atmospheric conditions or periods of rainfall. Again, the
quantity of comparison data was not sufficient to confirm any statistical correlation.
Loss of aerosols in the air intake systems can also complicate the derivation of $^{222}$Rn activity
concentrations from one-filter systems such as the HRM. Levin et al. (2017) carried out an assessment of
$^{222}$Rn progeny loss in long tubing by laboratory and field experiments. Results of these experiments, for
8.2 mm inner diameter (ID) Decabon tubing, gave an empirical correction function for $^{222}$Rn progeny
measurements, which enables the correction of measurements for this specific experimental setup (e.g.
tubing type and diameter, flow rate, aerosol size distribution).
Finally, Schmithüsen et al. (2017) conducted an extensive European-wide $^{222}$Rn/$^{222}$Rn progeny
comparison study in order to evaluate the comparative performance of one-filter and two-filter
measurement systems, determining potential systematic biases between them, and estimating correction
factors that could be applied to harmonize $^{222}$Rn activity concentration estimates for their use as a tracer
in various atmospheric applications. In this case, the authors employed a HRM monitor as the reference
device. It was taken to nine European measurement stations to run for at least one month at each of them.
This monitor was run in parallel to other one-filter and two-filter radon monitors operating at each station
of interest.
Although several inter-comparison campaigns have been carried out in the past, none of them has
included simultaneous observations from one-filter, two-filter and electrostatic deposition methods. Here,
we present the results of a three-month inter-comparison campaign carried out in the fall and winter of
2016-2017 in Gif Sur Yvette (France) where, for the first time, co-located measurements from monitors
based on the three measurement principles were included. Two two-filter $^{222}$Rn monitors, two single-filter
$^{222}$Rn progeny monitors and an electrodeposition monitor were run simultaneously under different
meteorological and aerosol conditions sampling from heights of 2 and 100 m a.g.l.
The main objectives of the present study were to: i) compare the calibration and correction factors
between all monitors required to derive harmonized atmospheric radon activity concentrations; and ii)
analyze the influence that meteorological and environmental parameters, as well as sampling height, can
have on the finally determined $^{222}$Rn activity concentration.
In the present manuscript the applied methodology is reported, including a short presentation of the $^{222}$Rn
/$^{222}$Rn progeny monitors participating in the campaigns, the sampling sites and the statistical analysis
carried out.  Finally, the outcomes of the present study are discussed and compared with the ones from
Schmithüsen et al. (2017).
**2 Methods**
In section 2.1 a short description is given of the monitors compared in the experiment, mainly focusing on
measurement techniques, instrument calibration and maintenance. The main characteristics of these
monitors are then summarized in Table 1. Section 2.2 presents the French atmospheric stations of Orme
de Mérisiers (ODM) and Saclay (SAC) where the two phases of the inter-comparison campaign were
realized. Section 2.3 briefly describes the devices used to measure the environmental parameters and the
atmospheric aerosol concentration at the above sites during the experiments. Finally, the statistical
analysis applied is described in section 2.4.
**2.1 $^{222}$Rn and $^{222}$Rn progeny monitors**
**2.1.1 Direct methods**
**Dual-flow-loop two-filter detectors**
The two 1500 L dual-flow-loop two-filter detectors included in this exercise were designed and built at
the Australian Nuclear Science and Technology Organisation (ANSTO). This model of detector, which
will henceforth be named ANSTO, is based on a previous design by Thomas and Leclare (1970), with
some early iterations of the modified design being described by Whittlestone and Zahorowski (1998) and
Brunke et al. (2002). The subsequent evolution of two-filter detectors in recent decades, and the current
principle of operation, has been described in detail by Williams and Chambers (2016) and Griffiths et al.
149    (2016).

During the measurement campaign ambient air was sampled continuously at a rate of about 83 L min$^{-1}$
through a 50 mm ID HDPE inlet tube and a 400 L delay volume to allow decay of the short-lived $^{220}$Rn
($T_{1/2}$= 56 s). The air stream then passes through the first filter, which removes all ambient aerosols as well
as $^{222}$Rn and $^{220}$Rn progeny. The filtered sample, now containing only aerosol-free air and $^{222}$Rn gas,
enters the main delay volume (1500 L) where $^{222}$Rn decay produces new progeny. The newly formed
$^{218}$Po and $^{214}$Po are then collected on a second filter and their subsequent α decays are counted with a ZnS
photomultiplier system. Atmospheric $^{222}$Rn activity concentrations are then calculated from the α count
rate and the flow rate through the chamber.
The detection limit ($L_D$) of two-filter detectors is directly related to the volume of the main delay
chamber. Here, $L_D$ is understood to represent the ambient radon concentration at which the estimated
counting error of the instrument reaches 30%. The $L_D$ of the 1500 L model used in this study was around
0.03 Bq m$^{-3}$. Under normal operation ANSTO monitors are automatically calibrated in situ every month
by injecting radon into the sampling air stream from a well-characterized Pylon $^{226}$Ra source (ca. 41 kBq
radium at SAC station) for 5 hours at a fixed flow rate of ~100 cc min$^{-1}$. Automatic instrumental
background checks, each lasting 24 hours, are also performed every 3 months to keep track of long-lived
$^{210}$Pb accumulation on the detectors second filter (which should be changed every 5 years). Based on a
calibration source uncertainty of 4%, coefficient of variability of valid monthly calibrations of 2-6%, and
a counting uncertainty of around 2% for radon concentrations ≥1 Bq m$^{-3}$, the total measurement
uncertainty of 1500 L ANSTO radon detectors is typically between 8% and 12% (k = 2). The ANSTO
monitors have low-maintenance requirements but, due to their dimensions (2.5 – 3m long) it can be
challenging to install them at stations with space restrictions. As an alternative to the 1500 L detectors, a
700 L model is also available, which is more portable and has a $L_D$ of 0.04-0.05 Bq m$^{-3}$. The combination
of detector volume, operating flow rate, and radon decay chain result in ANSTO monitors having a
response time of ~45 minutes, which can be corrected for in post processing (Griffiths et al. 2016).
Two ANSTO monitors were used during this study. As explained later in the text these monitors are
permanently running at SAC and ODM stations. No calibration source was available when the ANSTO
monitor was installed at the ODM site, so calibration and background information derived prior to
transport have been used.
**Electrostatic deposition monitor**
The Atmospheric Radon Monitor (ARMON) used in this experiment was designed and built at the Institut
de Tècniques Energètiques (INTE) of the UPC. The ARMON is a portable instrument based on
electrostatic deposition method, consisting of alpha spectrometry of positive ions of [218]Po electrostatically
collected on a detector (Hopke, 1989; Pereira and da Silva, 1989; Tositti et al., 2002). The ARMON is
described in detail in Grossi et al. (2012).
Sampled air with a flow rate between 1-2 L min$^{-1}$, is first filtered to remove ambient [222]Rn and [220]Rn
progeny and then pumped through a ~20 L spherical detection volume uniformly covered internally with
silver. Within this volume the newly formed [222]Rn and [220]Rn progeny, i.e. positive [218]Po and [216]Po ions,
respectively, are electrostatically collected on a Passivated Implanted Planar Silicon (PIPS) detector
surface by an electrostatic field inside the spherical volume. An 8 kV potential is applied between the
PIPS detector base and the sphere walls. As for the ANSTO detector, the sensitivity of this instrument
type depends on the detector volume. The design of the monitor employed in this study has a $L_D$ of about
0.07 Bq m$^{-3}$ in agreement with definition given above. Grossi et al., (2012) reported a minimum detection
limit for this instrument of around 0.2 Bq m$^{-3}$ in agreement with the definition of Gilmore, (2008). The
measurement efficiency of the electrodeposition method is reduced due to neutralization of the positive
[218]Po in recombination with OH$^-$ ions in the sampled air (Hopke, 1989). Consequently, it is necessary to
dry the sampled air as much as possible before it enters the detection volume. To this end, a dew point of
< -40°C was maintained at both inter-comparison sites using a cryocooler, consisting of a vessel tube
where sampling air was passing through before reaching the radon monitor (Grossi et al., 2018).
Each ARMON is calibrated at the INTE-UPC [222]Rn chamber (Vargas et al., 2004) under different [222]Rn
and relative humidity conditions (Grossi et al., 2012). The radon chamber of the INTE-UPC is a 20 m$^3$
installation, which allows control of the exhalation rate (0-256 Bq min$^{-1}$) and the ventilation air flow rate
(0-100 L min$^{-1}$). The [222]Rn source is a dry powder material containing 2100 kBq [226]Ra activity enclosed in
the source container (RN-1025 model manufactured by Pylon Electronics). The calibration factor $F_{cal}$ of
the ARMON used in this study was of 0.39 counts per minute (cpm) per Bq m$^{-3}$ with an uncertainty of
10% (k=2). The correction factor for the humidity influence inside the sphere was of 6.5·10$^{-5}$ per part per
million H$_2$O (ppm) with a maximum uncertainty of 10% (k=2). The total uncertainty of the atmospheric
radon activity concentration measured by the ARMON is of about 20% (k=2) for atmospheric [222]Rn
levels in the range of a few hundred Bq m$^{-3}$ but could increase up to 30-35% (k = 2) when atmospheric
[222]Rn levels decrease to a few Bq m$^{-3}$ due to the increase of the error of the alpha counts.  The total
uncertainty  includes the calibration factor $F_{cal}$, the background due to the presence of [212]Po from [220]Rn,
the net [218]Po counts and the humidity correction factor (Grossi et al., 2012; Vargas et al., 2015). Every 1-2
years the progeny filter at the ARMON inlet should be changed. The detection volume of the ARMON is
safety isolated because it is located within an external wooden cube of 0.18 m$^3$.

### 2.1.2 Non direct methods

### One-filter monitors

One-filter detectors measure the decay rates of aerosol-bound [222]Rn progeny directly accumulated by air
filtration (Schmithüsen et al., 2017). The [222]Rn activity concentration is then calculated assuming a
constant disequilibrium factor ($F_{eq}$) for a given site and sampling height between [222]Rn and the measured
progeny in the sampled air.
In the present study two monitors based on this method were used. One, named here as HRM, was
developed at the Institute of Environmental Physics of Heidelberg University, Germany, and is described
in detail by Levin et al. (2002). Rosenfeld (2010) describe the most recent version of this monitor for
which the electronics, data acquisition, and evaluation hardware and software were modernized. The
HRM measurement is based on α spectrometry of $^{222}$Rn daughters attached to atmospheric aerosols
collected on a static quartz fiber filter (QMA Ø 47 mm) using a surface barrier detector (Canberra CAM
900 mm$^2$ active surface). The $L_D$ of the HRM is about 0.07 Bq m$^{-3}$ at a flow rate of about 20 L min$^{-1}$ with
an uncertainty smaller than 15% (k=2) for atmospheric $^{222}$Rn levels above 2 Bq m$^{-3}$. This includes the
uncertainty of the line loss correction (see below) . Since one-filter detectors have no need for any delay
chambers but use only a compact filter holder with integrated detector and pre-amplifier, the HRM is a
small instrument with high portability. Regarding maintenance requirements, the quartz fiber filter should
be changed monthly.
During the measurement campaign carried out at the Saclay station, where air samples were collected via
a 100m Decabon tubing (see below), the line loss correction of Levin et al. (2017) was applied to all data
of the HRM. No loss of aerosol was assumed in the short tubing used at Orme de Mérisiers station. Here
we report for both sites $^{214}$Po activity concentrations. However, for the 100 m intake height at Saclay we
would not expect any disequilibrium, meaning that, based on the results from Schmithüsen et al. (2017),
the reported $^{214}$Po activity concentrations directly correspond to $^{222}$Rn activity concentrations. By contrast,
for the 2 m intake height at ODM we expect a $^{214}$Po/$^{222}$Rn disequilibrium of about 0.85 to 0.9.
The second type of one-filter monitor participating in this study was built at the Laboratoire des Sciences
du Climat et de l'Environnement, LSCE, France (Polian, 1986; Biraud, 2000; Schmithüsen et al., 2017).
Within this manuscript this monitor will be called the LSCE monitor. This monitor uses a moving filter
band system, which allows the determination of atmospheric $^{222}$Rn activity concentration based on
measurements of its progeny $^{218}$Po and $^{214}$Po. Attached $^{222}$Rn progeny are collected on a cellulose filter
(Pöllman–Schneider) over a one-hour period at a flow rate of 160 L min$^{-1}$ and after this aerosol sampling
period, the loaded filter is moved to the α spectrometry for a one hour measurement period by a
scintillator from Harshaw Company and photomultiplier from EMI, Electronics Ltd (Biraud, 2000). The
$L_D$ is about 0.01 Bq m$^{-3}$ with an uncertainty of about 20%.
Regarding maintenance on regular basis, the LSCE monitor's filter roll has to be changed every three
weeks. Automatic detector background is performed every three weeks and counting efficiency is
manually tested with an americium source. The instrument is designed to measure radioactive aerosols a
few meters above the ground level.  An inlet filter is installed to block black carbon or dirt deposition
when the instrument is installed in urban areas as the flow rate drops below 9 m$^3$ h$^{-1}$. The instrument size
is about 25 cm high, 40 cm long and 25 cm deep, and it can be easily deployed at a station.

| Monitor | Method | Sampling Flow Rate (L min⁻¹) | L_D (Bq m⁻³) | Typical uncertainty (k=2) | Portability considerations Dimensions (cmxcmxcm) and weight (kg) | Deployability | References |
|---|---|---|---|---|---|---|---|
| ANSTO | Dual-flow-loop two-filter | ~83 | ~0.03 | < 12% | 300x80x80 ~120 | • Remote control<br>• Time response correction<br>• Need of large pump if the simple intake line is more than ~10m in length | Whittlestone and Zahorowski (1998) ; Brunke et al. (2002) ; Chambers et al. (2018) |
| ARMON | Electrostatic deposition | ~2 | ~0.07 | < 35% | 90x80x80 ~10 | • α Spectrum<br>• Remote control<br>• Need of dry air simple | Grossi et al. (2012); Vargas et al. (2015) |
| HRM | One-filter | ~20 | ~0.07 | < 15% | 35x30x15 ~8 | • α Spectrum<br>• Remote control<br>• Sampling inlet height correction<br>• | Levin et al. (2002) |
| LSCE | One-filter | ~160 | ~0.01 | < 20% | 25x25x40 ~8 | • α Spectrum<br>• Remote control<br>• Sampling inlet height correction<br>• Need of large pump | Polian, (1986); Biraud, (2000) |

Table 1. Summary of principal characteristics of the $^{222}$Rn and $^{222}$Rn progeny monitors compared in the
present study.

## 2.2 Sites

The present inter-comparison study was carried out at two stations located 30 km southwest of Paris in
the fall and winter of 2016-2017 (Figure 1). Both stations, 3.5 km apart, belong to the LSCE and are
located in a region with a radon flux of ca. 5-10 mBq m$^{-2}$ s$^{-1}$ in winter, according to output of the Karsten
et al. (2015) model.
Phase I of the measurements started at Orme des Mérisiers (ODM, latitude 48.698, longitude 2.146, 167
m above sea level) and ran between 25 November 2016 and 23 January 2017. Here, LSCE and ANSTO
(for convenience named here as ANSTO_ODM) monitors are routinely running. During Phase I of the
inter-comparison exercise these two monitors were operated in parallel with a HRM and an ARMON.
The sampling height for all radon detectors at ODM was 2 m ag.l.
Phase II of the exercise was realized at Saclay (SAC, latitude 48.730, longitude 2.180, Figure 1) between
25 January 2017 and 13 February 2017. At this location the sampling inlet height was at 100 m a.g.l. At
SAC station another ANSTO monitor (from now on labelled as ANSTO_SAC) was already running. In
addition, during Phase II this detector was running in parallel with the portable ARMON and HRM
detectors. The LSCE monitor did not participate in Phase II of the experiment.
Meteorological parameters were also available at both stations during the inter-comparison periods at
heights corresponding to the radon measurements (2 m and 100 m a.g.l.). In the case of the ODM site,
atmospheric aerosol concentrations were also measured for this period.

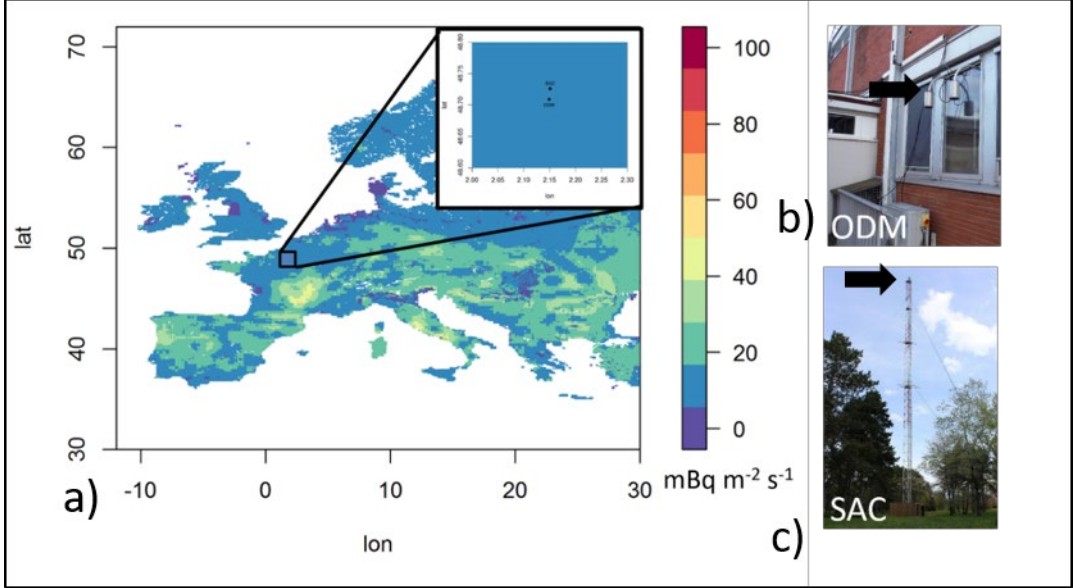

Figure 1. The INGOSv2.0 $^{222}$Rn flux map (Karstens et al., 2015) is shown for a typical winter month
(December), with locations of the ODM and SAC sites shown in inset (a). The radon sampling inlets are
indicated both for ODM (b) and SAC (c) by the black arrows.

**2.3 Environmental parameters and atmospheric aerosol concentration**

Meteorological data used within this study were available from continuous measurements carried out at
the SAC and ODM stations at 100 m and at 10 m a.g.l. respectively. The measurements were performed
with a Vaisala Weather Transmitter WXT520 (Campbell Scientific) for: (1) wind speed and direction
(accuracies of ± 3 % and ± 3 ºC, respectively); (2) Humidity and temperature (accuracies of ± 3 % and ±
0.3 ºC, respectively). In addition, the atmospheric aerosol concentration was measured at ODM site using
a fine dust measurement device Fidas® 200 S (Palas) at 10 m a.g.l.. The measurement range is between 0
and 20.000 particles cm$^{-3}$. All the accuracies refer to the manufacturer's specifications.

**2.4 Data Analysis**

**2.4.1 Correlation factors between monitors**

To study the correlation between responses of the different detectors, linear regression models were
calculated using hourly atmospheric radon activity concentrations from each monitor. The linear
regression fits were calculated following Krystek and Anton (2007), relative to the two portable detectors,
ARMON and HRM, because they both were measuring at SAC and at ODM.

**2.4.2 Analysis of the influence of the environmental and meteorological parameters on detector**
**response**

The present study intended to build upon the findings of Xia et al. (2010) and Schmithüsen et al., (2017)
regarding the possible influence of meteorological conditions on the response of radon and radon progeny
monitors.
With this in mind, the ratio between hourly atmospheric $^{222}$Rn activity concentrations measured and/or
obtained by the HRM, LSCE and ANSTO monitors, and that measured by the ARMON were calculated,
and their variability analyzed in relation to hourly atmospheric temperature, relative humidity and
atmospheric aerosol concentration measured at ODM and at SAC, respectively. Not enough rain data
were available to be used in this study. For this part of the study, the ARMON was used as reference
being the only direct radon monitor running at both sites.
**3 Results**
Hourly time serie s of atmospheric $^{222}$Rn, in the case of ARMON and ANSTO monitors, and $^{222}$Rn
progeny ($^{214}$Po activity concentration) for the HRM and LSCE monitors, measured at ODM and SAC
during Phase I and Phase II of the inter-comparison experiment are presented in Figures 2 and 3,
respectively. In each of the previous Figures, a zoom plot has been also reported as example to look at the
response of each monitor to the sub-diurnal atmospheric radon variability. As shown, all monitors
running at both sites follow this variability, with $^{222}$Rn and $^{222}$Rn progeny data measured or estimated by
the three different measurement techniques showing the same general patterns. Table 2 summaries the
means, minima and maxima hourly atmospheric radon or radon progeny activity concentrations measured
by each monitor for both campaigns.  For further information, Figures S1 and S2 of the supplementary
material show the time series of the differences (absolute) and of the ratios (relative) between the hourly
$^{214}$Po or $^{222}$Rn activity concentrations measured by HRM, LSCE and ANSTO monitors and those
measured by the ARMON.
**3.1 Phase I: ODM site**
During Phase I the LSCE, HRM, ARMON and ANSTO_ODM monitors were operating in parallel,
sampling air from the same height (2 m a.g.l.). The mean temperature over Phase I of the campaign was
2.9 ºC with an interquartile range of 0.10 ºC to 5.8 ºC. The mean relative humidity was 80% with an
interquartile range of 73% to 89%. An average accumulated rain per day of 13 mm was recorded. The
main wind patterns during Phase I were from northeast and southwest, with speeds typically between 1
and 7 m s$^{-1}$. The mean atmospheric aerosol concentration observed at ODM during Phase I was 505
particles cm$^{-3}$ with an interquartile range of 233 cm$^{-3}$ to 660 cm$^{-3}$.
The means of the atmospheric $^{222}$Rn activity concentration measured by the ARMON and the
ANSTO_ODM are in the same order (Table 2). The means of the atmospheric $^{214}$Po activity
concentrations measured by LSCE monitor were ca. 50% lower and by the HRM ca. 30% lower than the
atmospheric $^{222}$Rn activity concentration.

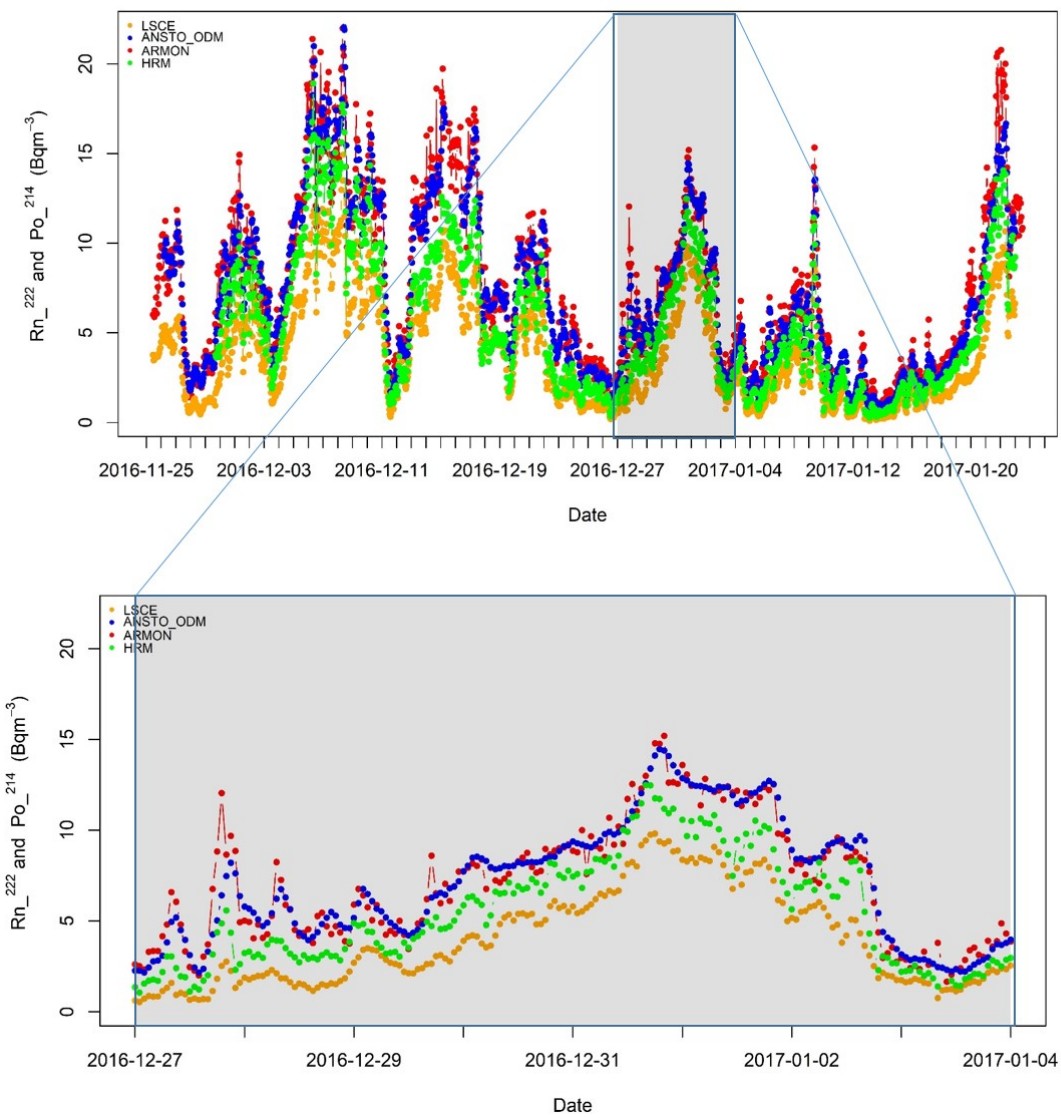

Figure 2. Main panel: Hourly time series of the atmospheric $^{222}$Rn and, in the case of LSCE and HRM data $^{214}$Po activity concentration, measured at Orme de Merisiers (ODM) station during Phase I (between 25 November 2016 and 23 January 2017) by: ARMON (red circles), ANSTO_ODM (blue circles), HRM (green circles) and LSCE (orange circles) monitors. Zoomed panel: Hourly time series of the atmospheric $^{222}$Rn and $^{214}$Po measured between 27[th] December 2016 and 04[th] January 2017.

Table 2 shows the slopes (*b*) and intercepts (*a*) of the linear regression fits calculated between the hourly atmospheric $^{222}$Rn and $^{214}$Po activity concentrations measured by the ARMON and/or the HRM and the other $^{222}$Rn and $^{222}$Rn progeny monitors deployed in Phase I. The calculated slopes were in the range of 0.62 to 1.17 and the $R^2$ values varied between 0.90 and 0.96. The slope closest to unity was calculated between the ARMON and ANSTO_ODM monitors, and was 0.96±0.01, while the lowest slope was observed between the ARMON and LSCE monitors, and was 0.62±0.01. The highest correlation ($R^2$=0.96) was found between the HRM and LSCE monitors. The plots of the linear regression fits of the Phase I are shown in the left panels of the Figures S3, S4 and S5 of the supplementary material. Notably,

the offset (*a* value) of the regression between the ANSTO and ARMON detectors at ODM is considerably greater than that at SAC. The regression slopes are also slightly different. These differences are likely related to the limited calibration and background information available for the ANSTO_ODM detector for this inter-comparison project. In particular, a substantial component of the instrumental background signal is site specific. This is likely responsible for much of the change in offset value.

| | | | | | | | x | | |
|---|---|---|---|---|---|---|---|---|---|
| | Monitors Phase I | Mean (Bq m⁻³) | Min/Max (Bq m⁻³) | *b* (ARMON) | *a* (ARMON) | R² (ARMON) | *b* (HRM) | *a* (HRM) | R² (HRM) |
| | ANSTO_ODM | 7.02 | 0.73/22.04 | 0.96±0.01 | -0.23±0.03 | 0.94 | 1.17±0.01 | 0.63±0.03 | 0.93 |
| | HRM | 5.45 | 0.26/18.91 | 0.82±0.01 | -0.71±0.03 | 0.93 | - | - | - |
| | ARMON | 7.55 | 0.50/21.98 | - | - | - | - | - | - |
| y | LSCE | 3.84 | 0.10/14.93 | 0.62±0.01 | -0.85±0.03 | 0.90 | 0.76±0.004 | -0.29±0.03 | 0.96 |
| | Monitors Phase II | Mean (Bq m⁻³) | Min/Max (Bq m⁻³) | Slope (ARMON) | Intercept (ARMON) | R² (ARMON) | Slope (HRM) | Intercept (HRM) | R² (HRM) |
| | ANSTO_SAC | 3.50 | 0.43/10.71 | 0.97±0.01 | 0.01±0.06 | 0.95 | 1.03±0.01 | 0.15±0.06 | 0.90 |
| | HRM | 3.26 | 0.26/11.15 | 0.94±0.01 | -0.13±0.06 | 0.91 | - | - | - |
| | ARMON | 3.60 | 0.17/11.51 | - | - | - | - | - | - |

Table 2. The means, maxima, and minima of the atmospheric $^{222}$Rn and $^{214}$Po activity concentration observed by each monitor participating in the Phase I and II of the inter-comparison campaigns. The slopes (*b*) and intercepts (*a*) of the linear regression fits calculated between the hourly atmospheric $^{222}$Rn and $^{214}$Po activity concentrations measured by the ARMON and/or the HRM and the other $^{222}$Rn and $^{222}$Rn progeny monitors deployed in both phases are also reported.

**3.2 Phase II: SAC station**

Phase II lasted 18 days. The mean temperature during this period was 5 ºC with an interquartile range of 2 ºC to 8 ºC. The mean relative humidity was 86% with an interquartile range of 80% to 94%. An average accumulated rain per day of 3 mm was recorded. The main wind patterns during this phase at 100 m a.g.l. were from the south and southwest with speeds typically between 3 and 10 m s⁻¹.

Figure 3 shows the hourly atmospheric $^{222}$Rn and $^{214}$Po activity concentrations observed at SAC during Phase II by the ARMON, HRM and ANSTO_SAC instruments.

Table 2 reports the means, minima, and maxima of the atmospheric data measured during Phase II by all participating monitors. In this case, the mean atmospheric $^{222}$Rn and $^{214}$Po activity concentrations measured by all monitors agreed within the instrumental errors. At 100 m a.g.l. the slopes of the hourly fits of the monitor's response in this case were all close to unity. The calculated offsets also decreased at 100 m a.g.l. relative to 2 m a.g.l.  The plots of the linear regression fits of Phase II are shown in the right panel of Figures S5 and S6 of the supplementary material. During the period of Jan 30 – February 1, 2019, the HRM shows significantly lower values than the ANSTO and ARMON. This period coincides with saturated air humidity conditions.

Figure S7 of the supplementary material presents two plots to summarize the results of the slopes and offsets calculated both at ODM and SAC stations relative to the ARMON.

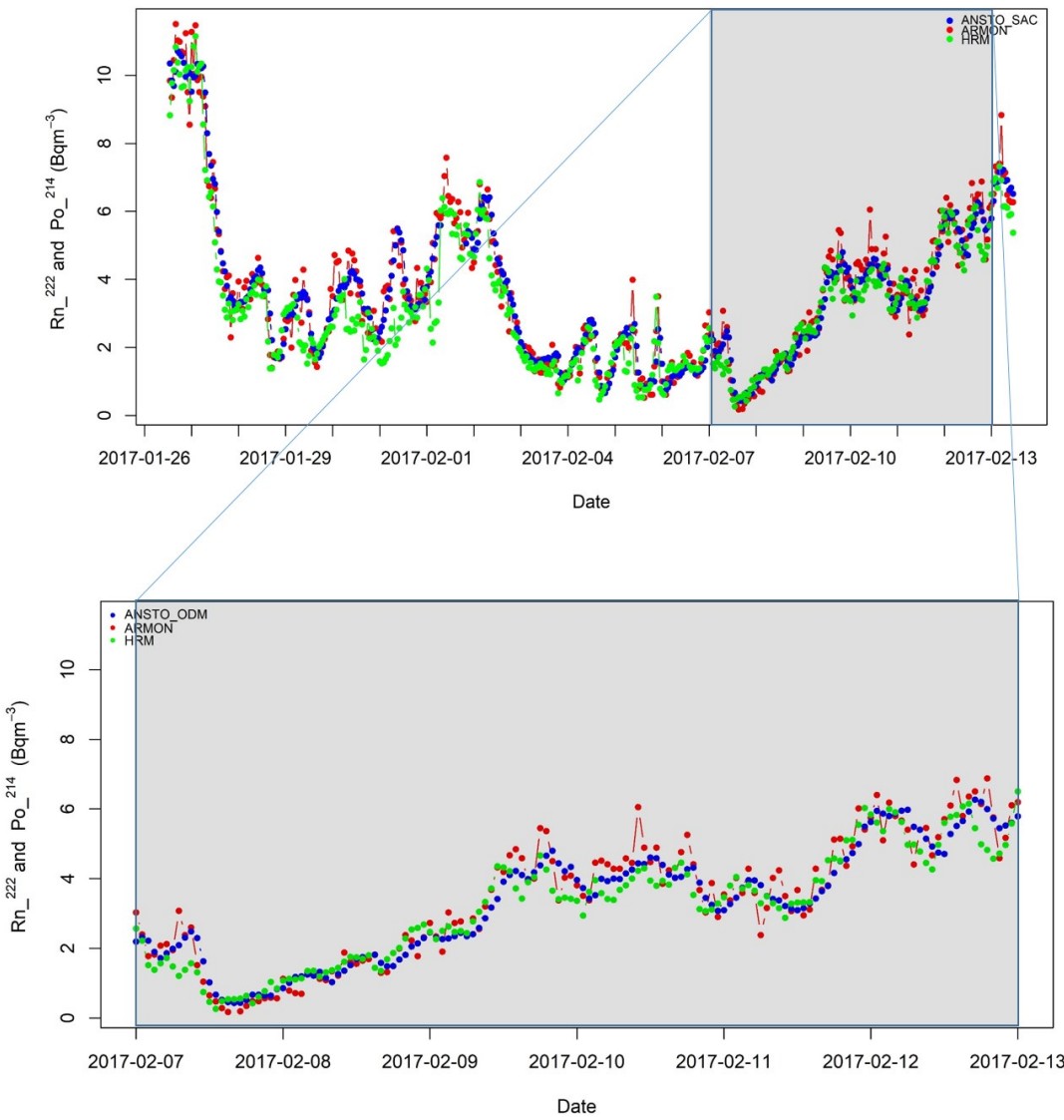

Figure 3. Main panel: Hourly time series of the atmospheric $^{222}$Rn and $^{214}$Po (HRM) activity concentration measured at Saclay (SAC) station between 25 January 2017 and 13 February 2017 by: ARMON (red circles), ANSTO_SAC (blue circles) and HRM (green circles) monitors. Zoomed panel: Hourly time series of the atmospheric $^{222}$Rn and $^{214}$Po measured between 7 February 2017 and 13 February 2017.

Figure 2 and 3 show a larger hourly variability of the HRM and ARMON signals compared with the ANSTO ones. This difference in variability is likely due to a larger uncertainty of the HRM and ARMON detectors for atmospheric $^{222}$Rn levels of around 1 Bq m$^{-3}$. In addition, it has to be taken into account that only an approximated form of the Griffiths et al. (2016) response time correction could be applied to the output of the ANSTO detectors. Further investigations should be carried out to clarify these differences and to exactly quantify the detectors uncertainties for the low $^{222}$Rn concentrations typical for outdoor environmental monitoring at or above 100 m a.g.l.

## 3.2 Comparison with past studies

The results obtained in the present study of the slopes (b) and of the offsets (a) of the regression lines calculated between ANSTO or LSCE monitors against the HRM are here compared with the ones presented by Schmithüsen et. al., 2017. Table 3 shows a summary of this comparison. All slopes (correction factors) are defined as (routine station monitor) / HRM because this last was used as reference instrument by Schmithüsen et. al., 2017.

| Site/Input Height | Schmithüsen et al., 2017 | | | Present study | | |
|---|---|---|---|---|---|---|
| **ANSTO/HRM** | **Activity Range (Bq m$^{-3}$)** | **b** | **a** | **Activity Range (Bq m$^{-3}$)** | **b** | **a** |
| Cabauw: 200/180 m | 0-8 | 1.11±0.04 | 0.11±0.06 | | | |
| Saclay: 100 m | | | | 0-11 | 1.03±0.01 | 0.15±0.06 |
| Lutjewad: 60 m | 0-6 | 1.11 ± 0.02 | 0.11 ± 0.02 | | | |
| Heidelberg: 35 m | 0-15 | 1.22 ± 0.01 | 0.42 ± 0.04 | | | |
| Cabauw: 20 m | 0-12 | 1.30 ± 0.01 | 0.21 ± 0.03 | | | |
| Orme des Mérisiers: 2 m | | | | 0-22 | 1.17±0.01 | 0.63±0.03 |
| **LSCE/HRM** | **Activity Range (Bq m$^{-3}$)** | **b** | **a** | **Activity Range (Bq m$^{-3}$)** | **b** | **a** |
| Orme des Mérisiers: 2 m | 0-9 | 0.68±0.03 | -0.18±0.09 | 0-15 | 0.76±0.01 | -0.29±0.03 |

Table 3. Offsets and slopes of the regression lines calculated between ANSTO or LSCE monitors against the HRM in the present study and by Schmithüsen et. al., 2017.

Data in Table 3 need to be analysed taking into account that a unique traceability chain is not yet available for atmospheric radon measurements and the different monitors routinely running at the different stations could have different calibration chains (e.g. radon source, primary standard, etc.). Generally speaking, for both studies, it can be observed that the correction factor between the atmospheric $^{214}$Po activity concentration measured by HRM and the atmospheric $^{222}$Rn activity concentration measured by ANSTO at each station approaches unity with the increase of the height of the sampling input. By contrast, the offsets of the regression fits decrease with the increase of the input height.

The only case where the compared instruments were exactly the same and at the same height is for Orme des Mérisiers station. Here the slope between the atmospheric $^{214}$Po activity concentration measured by LSCE and HRM is equal to 0.76±0.01. This number is slightly larger but within uncertainties well comparable to the number reported by Schmithüsen et al. (2017) of 0.68±0.03 (see Table 3).

### 3.4 Influence of the weather conditions on the ratio between $^{214}$Po and $^{222}$Rn measurements

Figure 4 shows the variability of the ratio between hourly atmospheric $^{214}$Po and/or $^{222}$Rn activity concentration measured by each monitor relative to those measured by the ARMON versus the hourly means of ambient temperature and relative humidity. Analysis was carried out at ODM (Figure 4, upper panels) and at SAC (Figure 4, bottom panels) versus ambient temperature (Figures 4, left panels) and relative humidity (Figures 4, right panels) measured at the corresponding stations.

Figure 5 shows the same variability plotted in relation to the ANSTO_ODM at ODM (Figure 5, upper panels) and to the ANSTO_SAC at SAC (Figure 5, bottom panels) versus the hourly means of ambient temperature (Figures 5, left panels) and relative humidity (Figures 5, right panels).

Data does not show any evident patterns at 100 m a.g.l. (SAC station), which could indicate that there is any impact on $^{222}$Rn or $^{222}$Rn progeny measurements due to change of ambient temperature and relative humidity, at least not until saturated conditions are achieved. By contrast, a small decrease, of about 10$^{-2}$

ºC⁻¹, is observed in the ratio between the ²¹⁴Po activity concentration (measured by HRM and LSCE monitors) and the ²²²Rn activity concentration (measured by ANSTO_ODM and ARMON monitors) with the increase of the ambient temperature (Figure S8 of the supplementary material) at 2 m a.g.l. (ODM station). This temperature dependency may be rather due to the effect of atmospheric activity concentrations, increasing during nightime, on the disequilibrium between radon and its progeny. However, this influence on measured ²¹⁴Po/²²²Rn ratios is really small compared with others observed effects (e.g.: loss of progeny within the sample tube (Levin et al., (2017)), atmospheric aerosol concentration (see below)). Looking at Figure 5, there appears to be less scatter in the point clouds (particularly at SAC) when the ANSTO_SAC monitor is used as the reference, likely attributable to the lower measurement uncertainty of the ANSTO monitor used at this station.

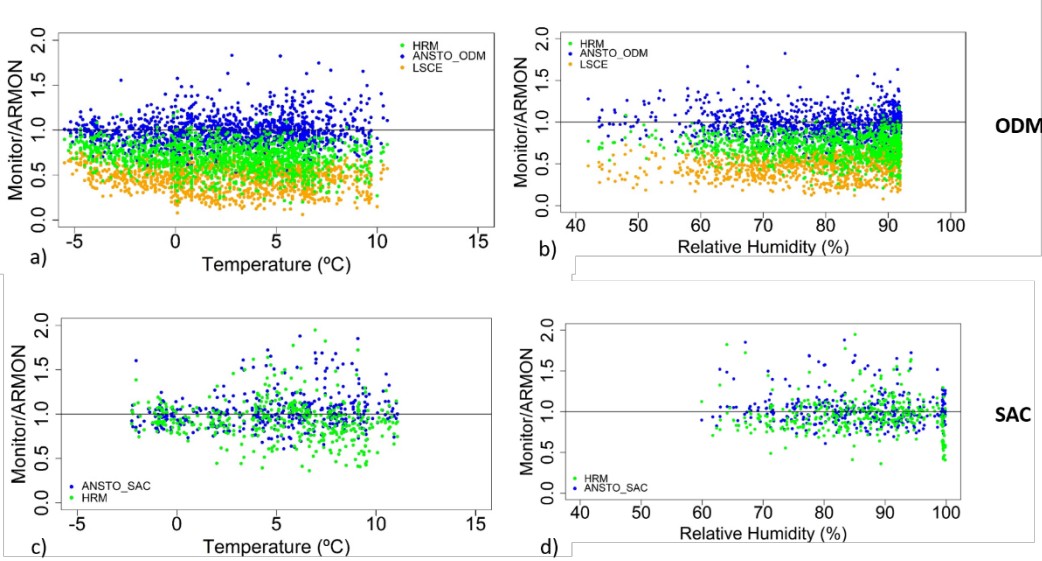

Figure 4. Hourly atmospheric ²²²Rn or ²¹⁴Po activity concentration obtained by HRM, LSCE and ANSTO monitors divided by the ²²²Rn activity concentration measured by the ARMON detector as function of the hourly measured atmospheric temperature and relative humidity at ODM (a and b) and at SAC (c and d), respectively.

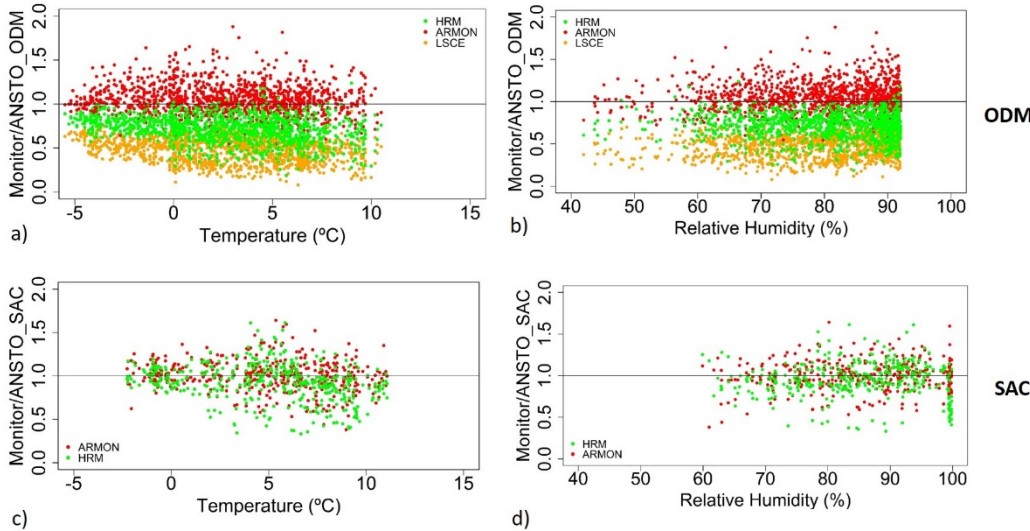


Figure 5. Hourly atmospheric $^{222}$Rn or $^{214}$Po activity concentration obtained by ARMON, HRM and
LSCE monitors divided by the $^{222}$Rn activity concentration measured by the ANSTO detectors as function
of the hourly measured atmospheric temperature and relative humidity at ODM (a and b) and at SAC (c
and d), respectively.
In Figure 6 the ratio of the hourly atmospheric $^{222}$Rn or $^{222}$Rn progeny activity concentration measured by
the HRM ($^{214}$Po in Figure 6a), the LSCE ($^{214}$Po in Figure 6b) and the ANSTO_ODM ($^{222}$Rn in Figure 6c)
monitor and the $^{222}$Rn activity concentration measured with ARMON ($^{222}$Rn) are plotted against the
logarithm of the hourly aerosol concentration data. Data indicate the existence of a linear relationship
between these variables, i.e. of the form:
$$\frac{^{222}Rn\,(Monitor\_i)}{^{222}Rn\,(ARMON)} = a + b \cdot Log_{10}(Aerosol\ Conc.). \hspace{2cm} (1)$$
Here *$^{222}$Rn (Monitor_i)* is the hourly atmospheric $^{222}$Rn or $^{214}$Po activity concentration measured by
individual monitors HRM ($^{214}$Po), LSCE ($^{214}$Po) and ANSTO_ODM ($^{222}$Rn), *$^{222}$Rn (ARMON)* is the one
measured by the ARMON monitor and *Aerosol Conc.* is the hourly atmospheric aerosol concentration
measured at ODM during Phase I. The results of the linear regression fits are reported in Table 4. The
slope of the ratio between the ANSTO_ODM and ARMON monitors in relation to the variability of the
logarithm of the hourly atmospheric aerosol concentration is close to zero and the intercept is close to
one. The ratio between the hourly atmospheric aerosol-bound radon progeny data measured by the two
one-filter radon progeny monitors and the one measured by the ARMON seems to decrease with
decreasing aerosol concentration (Figures 6a and 6b). However, this effect becomes only evident when
atmospheric aerosol concentration is lower than 300 particles cm$^3$.

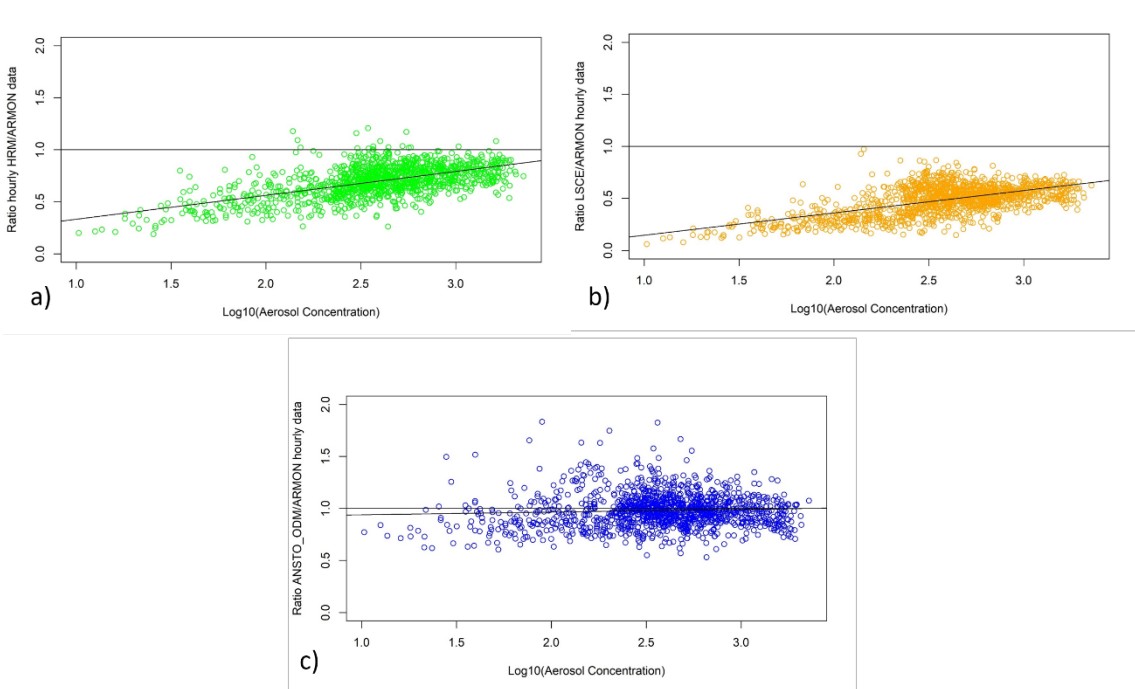


Figure 6. Ratio of the atmospheric $^{222}$Rn or $^{214}$Po activity concentration measured by the HRM (green dots), LSCE (orange dots) and ANSTO_ODM (blue dots) monitors and those measured by the reference ARMON monitor against the logarithm of the atmospheric aerosol concentration measured at ODM station.

| Monitor | $a$ | $b$ | $R^2$ |
|---------|-----|-----|-------|
| HRM | 0.10±0.02 | 0.23±0.01 | 0.34 |
| LSCE | -0.07±0.02 | 0.21±0.01 | 0.34 |
| ANSTO_ODM | 0.91±0.03 | 0.03±0.01 | $0.04 \cdot 10^{-1}$ |

457    Table 4. Intercepts and slopes of the linear regression fits of the Equation 1

**Conclusions**

In order to confirm and build upon the results obtained by Xia et al. (2010), Grossi et al. (2016) and Schmithüsen et al. (2017) a three month inter-comparison campaign was carried out in the south of Paris, France, in the fall-winter period of 2016-2017. For the first time, three fundamentally distinct radon and radon progeny measurement approaches were compared side-by-side at two measurement heights: 2 and 100 m a.g.l., under a range of environmental conditions with the aim to compare their responses.

The results of this study show that $^{222}$Rn and $^{222}$Rn progeny measurements follow the same general patterns of diurnal variability, both close to and further up from the surface. The slopes of the linear regression fits between the radon and the radon-progeny measurements, which represent the calibration factors, are not significantly different from one at 100m height above ground (SAC), but they differ at the 2 m level (ODM). The latter is attributable to the disequilibrium known to exist between $^{222}$Rn freshly emitted from the ground and its short-lived progeny in the lowest 10s of meters of the atmosphere, the magnitude of which is known to decrease with distance from the surface, as shown in earlier work, and to be close to one at a height of 100 m and above (e.g. Jacobi and André, 1963; Schmithüsen et al., 2017).

For the 2 m level, we found a significant correlation of radon progeny activity concentrations between
LSCE and HRM measurements (see Figure S3 in the Supplement). The slope, however, is only equal to
0.76±0.01. This result is comparable, considering its uncertainties, with the result  reported by
Schmithüsen et al. (2017) of 0.68±0.03 (see Table 3) based on the comparison of the same two monitors
(HRM and LSCE) and at the same station (ODM) in March and April 2014.
Observations of the total atmospheric aerosol concentration available at ODM station during the first two
months of the experiment were used to investigate the influence of changing atmospheric aerosol
concentrations on the response of the radon/radon progeny measurements. Under very low atmospheric
aerosol loading (< 300 particles cm$^{-3}$), the $^{222}$Rn progeny monitors seem to underestimate the atmospheric
$^{214}$Po activity concentrations by up to 50%. This effect may be attributable to loss of un-attached $^{218}$Po
and $^{214}$Po. Particle number concentrations below 300 particles cm$^{-3}$ at continental stations are, however,
very rare and even during winter at Alpine stations like Schneefernerhaus such low particle
concentrations are only occasionally observed (Birmili et al., 2009).
The comparison of results obtained in the present study with those reported in Schmithüsen et al. (2017)
demonstrate that in order to harmonize atmospheric $^{222}$Rn activity concentrations measured at different
atmospheric networks it will be important to: i) have a well-established metrological chain; ii) have
traceable methods for measuring low-level atmospheric radon activity concentrations; iii) harmonize the
calculation of total uncertainty in atmospheric $^{222}$Rn concentrations measured by all monitors when
ambient radon is only a few Bq m$^{-3}$ or less; iv) use a direct radon monitor as a mobile reference
instrument, the response of which is not influenced by meteorological conditions or inlet tube dimensions
and length.
Finally, the new portable ARMON seems to have a great potential for being used at atmospheric radon
stations with space restrictions. It could also be useful as mobile reference instrument to calibrate $^{222}$Rn
progeny instruments or fixed radon monitors. However, the total expanded uncertainty of the ARMON
could increase for really low radon activity concentration (<1 Bq m$^{-3}$) and when atmospheric $^{220}$Rn is also
present. This should be better investigated in the near future. As should being investigated the
uncertainties related to the ANSTO detector response time correction when characteristics of the entire
intake system have not been directly measured

**Acknowledgments**
The research leading to these results has received funding from the Ministerio Español de Economía y
Competividad, Retos 2013 (2014–2016) with the MIP (Methane interchange between soil and air over the
Iberian Península) project (reference: CGL2013-46186-R). This study was carried out under the umbrella
of the Atmospheric Thematic Center (ATC) of ICOS.
Claudia Grossi particularly thanks the Ministerio Español de Educación, Cultura y Deporte, for partially
supporting her work with the research mobility grant "José Castillejos" (ref. CAs15/00042).

The authors warmly thank (i) the INTE team, in the persons of Vicente Blasco and Juan Antonio Romero, for their work in the building of the ARMON used in this study; (ii) the R project ([www.r-project.org](www.r-project.org)) free software environment used here for statistical computing and graphics.

This paper is dedicated to: Bruno Grossi, Dr. Manuel Javier Navarro Angulo, Dr. Alfredo Adán and the whole team of the Instituto Clínic de Oftalmología (ICOF) of the Hospital Clínic of Barcelona.

**Code/Data availability**

The raw data and the R codes used for this study are available at: https://www.dropbox.com/sh/xokyu4vnt6f0gme/AABt-DxnTBbe6FFT9p4WDZWda?dl=0

**Author contribution**

C. Grossi, O. Llido, F. R. Vogel, V. Kazan, M. Delmotte, R. Curcoll, J.-A. Morguí, S. D. Chambers and A. Capuana worked at the installation of the radon and the radon progeny monitors. In addition, they were in charge of the maintenance of the in situ and remote radon and radon progeny measurements during the 3 months of experiment. C. Grossi, O. Llido, V. Kazan, D. Chambers, S. Werczynski, A. Capuana, I. Levin worked at the calculation and delivery of the radon and radon progeny time series data. M. Delmotte, M. Ramonet worked on the availability of the meteorological and aerosol data covering the inter-comparison campaign period.

All authors collaborate in the discussion on the data results and participate in the writing of the current manuscript.

**Competing interests**

All authors declare that they do not have competing interests related with the results of this study.

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
