# Peer review of "Inter-comparison study of atmospheric $^{222}\text{Rn}$ and $^{222}\text{Rn}$"

_Atmospheric Measurement Techniques, 2019_

## Referee Comment (RC1) · Grant Forster (Referee) · 2 Dec 2019

General comments

This study further highlights the problems faced when making measurements of 222Rn activity concentrations with different techniques and highlights some of the major issues which affect the comparability of the different measurement techniques (e.g. sampling height, aerosol loading, and atmospheric humidity). The study builds upon previous Inter-comparison studies between 222Rn and 222Rn progeny monitors (e.g. Schmithüsen et. al., 2017) through the introduction of a second "direct" measurement technique, the ARMON monitor. The agreement between the ANSTO and the ARMON monitors is very good at different sampling heights and in changing meteorological

conditions. This is good news for the atmospheric radon monitoring community due to the portability of the ARMON monitor when compared with the ANSTO monitor. However, the ANSTO monitor still outperforms the ARMON monitor and has a lower limit of detection and less noise as demonstrated in this manuscript. Therefore this must be taken in to account in very clean conditions. The manuscript highlights that there is a need for "a more statistically robust evaluation" of the discussed influences on the 222Rn activity concentration measurements and highlights that a longer dataset is needed. I feel that this brings the story up to date as efforts are increased across Europe to improve and expand the radon monitoring activities and this manuscript will act as an important steer to any decision making. I recommend that this manuscript is published with some minor revisions.

Specific comments

1. I feel that a solid aspect of this paper is that the ARMON monitor performs extremely well and has excellent potential for deployment in radon networks. The other instruments have all been components of previous inter-comparison studies. Therefore, I suggest that the manuscript should be ARMON centric rather than being an inter-comparison study. I think that there is huge value in the work presented herein and the ARMON should be showcased. Perhaps change the title of the manuscript to reflect this?

2. In the abstract, the author mentions that this paper evaluates "correction factors between monitors". I think that the author needs to highlight that the slopes from the scatter plots are the correction factors.

3. I would like to see a section which compares the outcomes of this study with those from previous instrument comparisons (e.g. Schmithüsen et. al., 2017) to put the findings into context. How well do they agree? How site-specific are these corrections and what can be done to overcome this? What needs to be considered in future inter-comparison studies?

4. I think it would help to rearrange the methods section to clearly state that "direct" and "non-direct" methods are being compared. As highlighted above I feel that this is the really strong part of the manuscript as this brings in a second "direct" measurement.

5. Section 2.1.2. Can you add a little bit of information to describe how the measured progeny from the HRM one-filter monitor is related to 222Rn activity concentration? This is discussed in Schmithüsen et al (2017) but it would be good to see it repeated here.

6. Section 2.1.3. It is stated that the ARMON is portable. Can you elaborate and possibly give the dimensions?

7. I suggest an additional figure with a synthesis of the slopes between the different monitors that are summarised in Table 2. This could be in the form of ANSTO vs. all of the other monitors for each site. However, keep table 2 as it contains all of the detail, it's just not easy to picture and visualise. I have added a figure to demonstrate what I mean.

Technical comments Figures: Sometimes hard to distinguish between the blue traces (ANSTO) and the black traces (ARMON) on the figures. However, this may be due to my eyes? Line 42: replace "because of the" with "from the". Lines 200 – 201: "method C". It's unclear what this means. Line 251 and 252: I don't understand this sentence. Line 251 - 257: Switched tense after the first sentence. Line 255: Replace "Fine" with "fine" Line 261: Replace "in order to" with "To" Line 353 – 358: This long sentence is hard to follow. Please revise. Line 383: Remove "compared" Use "$\alpha$" or "alpha" Use "progeny" or "daughters".

ANSTO
HRM
LSCE

**Fig. 1.**

---

## Referee Comment (RC2) · Susana Barbosa (Referee) · 12 Dec 2019

**General comments**

The manuscript describes the inter-comparison of three distinct instruments used for the measurement of atmospheric radon concentration. The topic is highly relevant given the diverse applications of radon as an atmospheric tracer and the clear benefit of documenting the performance of the most commonly used instruments for measurement of radon concentration in the air. The manuscript is clearly written and in my opinion scientifically sound. I only have some minor remarks detailed below.

**Specific comments**

Fig. 1: maybe add small arrows pointing to the inlets, particularly in case (c)

Section 2.3: the first sentence (lines 251-252) is not clear to me... I would also suggest specifying the height at which the meteorological measurements are taken, as well as the atmospheric aerosol concentration

Figure 2: possibly display also (maybe as supplemental material) the plot of the difference time series

Section 3.3: in my opinion it is not clear that data does not show any evident pattern... for example, at least by eye, seems to me that LSCE and HRM values relative to AR-MON as well as relative to ANSTO_ODM show a decreasing trend with temperature...

Page 16, line 421: maybe aerosol loading (instead of aerosol burden)

---

## Author Comment (AC1) · 16 Dec 2019

We want to thank Grant Forster for his review of the paper. The suggestions and comments have been considered for the improvement of the manuscript. In the following lines, the answers (in blue color) to his specific comments are developed. The recommended changes within the manuscript will be applied as soon as the open discussion will be ended.

**Specific comments**

1. I feel that a solid aspect of this paper is that the ARMON monitor performs extremely well and has excellent potential for deployment in radon networks. The other instruments have all been components of previous inter-comparison studies. Therefore, I suggest that the manuscript should be ARMON centric rather than being an inter-comparison study. I think that there is huge value in the work presented herein and the ARMON should be showcased. Perhaps change the title of the manuscript to reflect this?

We agree with the reviewer than the introduction of a new direct radon monitor, as the ARMON, in the inter-comparison of radon/radon progeny monitors for atmospheric concentration measurements is the solidest aspect of this work. We also agree that this monitor seems to have a great potential to be used within radon networks. The measurement technique of the ARMON is not new because it was already applied in previous instruments such as one built at the Brazilian National Institute for Space Research (INPE) (Pereira and da Silva, 1989; Tositti et al., 2002). In addition, the ARMON monitors have been already used in the past years for different studies in the atmospheric research field (Grossi et al., 2012; Vargas et al., 2015; Hernandez-Ceballo et al., 2015; Grossi et al., 2016; Grossi et al., 2018).

However we like to pointed out that, from a general point of view, it is the first time that four direct/indirect radon monitors, based on different measurement methods, have been running in parallel at the same site and at different heights. This gives the opportunity of comparing their responses under the same atmospheric and meteorological conditions. It is also the first time that the ARMON response is compared with the one of another direct radon monitor such as the ANSTO which is quite well characterized.

As correctly stated by the reviewer, the ARMON also shows a higher detection limit compared with the ANSTO and a larger uncertainty. At the same time seems that the ANSTO does a smoothing over the time series when fast changes in the atmospheric radon concentration are occurring. In order to correctly evaluate all these previous observations, the authors think that it is necessary, and are already planning, a long term inter-comparison campaign to specifically analyze the ARMON performances vs ANSTO's ones as explained also in the conclusion of this paper.

Therefore, we would like to present the results of these comparisons between different monitors without using any one as reference instrument neither focusing on anyone.

However, the modified version of our manuscript will showcase the introduction of a new portable direct radon monitor, the ARMON, its potential and the importance of completely evaluating its qualities and faults as direct radon monitor for atmospheric stations.

2. In the abstract, the author mentions that this paper evaluates "correction factors between monitors". I think that the author needs to highlight that the slopes from the scatter plots are the correction factors.

**It will be done as suggested by the reviewer.**

3. I would like to see a section which compares the outcomes of this study with those from previous instrument comparisons (e.g. Schmithüsen et. al., 2017) to put the findings into context.

How well do they agree? How site-specific are these corrections and what can be done to overcome this? What needs to be considered in future inter-comparison studies?

We agree that a section where the findings of this studies will be compared with the ones founded in previous studies could be of interest.

We decided here to compare the slope/offset of the regression lines calculated in this study between ANSTO and LSCE monitors against the HRM because they were also calculated in Schmithüsen et. al., 2017 for others ANSTO monitors and at different heights. The slopes (correction factors) are defined as (routine station monitor) / HRM.

However, looking at these data it is important to take into account that so far a unique traceability chain is not yet available for atmospheric radon measurements and the different monitors used at the different stations could have different calibration chains (ex. radon source, primary standard, etc.). It is also important to underline that the heights of the input was different at each station in Schmithüsen et. al., 2017 and this, as reported in Levin et al., 2017, implies the need of a correction on the 218Po concentration measured by the HRM, which was used as reference. In Schmithüsen et. al., 2017 these corrections were not applied.

| Site/Input Height              | Schmithüsen                                | Schmithüsen     | Schmithüsen
et al. 2017 | Present                                    | Present   | Present study |
|--------------------------------|--------------------------------------------|-----------------|----------------------------|--------------------------------------------|-----------|---------------|
| ANSTO monitors/HRM             | Activity
Range (Bq
m -3 ) | Slope           | Offset                     | Activity
Range
(Ba m -3 ) | Slope     | Offset        |
| Cabauw: 200/180 m              | 0-8                                        | 1.11±0.04       | 0.11±0.06                  | (bq m )                                    |           |               |
| Saclay: 100 m                  |                                            |                 |                            | 0-11                                       | 1.03±0.01 | 0.15±0.06     |
| Lutjewad: 60 m                 | 0-6                                        | $1.11 \pm 0.02$ | $0.11 \pm 0.02$            |                                            |           |               |
| Heidelberg: 35 m               | 0-15                                       | $1.22\pm0.01$   | $0.42\pm0.04$              |                                            |           |               |
| Cabauw: 20 m                   | 0-12                                       | $1.30\pm0.01$   | $0.21 \pm 0.03$            |                                            |           |               |
| Orme des Mérisiers: 2 m        |                                            |                 |                            | 0-22                                       | 1.17±0.01 | 0.63±0.03     |
| LSCE One-filter
Monitor/HRM |                                            |                 |                            |                                            |           |               |
| Orme des Mérisiers: 2 m        | 0-9                                        | 0.68±0.03       | -0.18±0.09                 | 0-15                                       | 0.76±0.01 | -0.29±0.03    |

The table below will be included and discussed in the modified version of our manuscript.

4. I think it would help to rearrange the methods section to clearly state that "direct" and "nondirect" methods are being compared. As highlighted above I feel that this is the really strong part of the manuscript as this brings in a second "direct" measurement.

We will rearrange the methods section as suggested by the reviewer and we will also underline the importance of the presence of a second direct radon monitor.

5. Section 2.1.2. Can you add a little bit of information to describe how the measured progeny from the HRM one-filter monitor is related to 222Rn activity concentration? This is discussed in Schmithüsen et al (2017) but it would be good to see it repeated here.

We will add this information in the text.

6. Section 2.1.3. It is stated that the ARMON is portable. Can you elaborate and possibly give the dimensions?

Sure, we will add this information in Table 1 in the "portability level" column (see below)

| Monitor | Method                                   | α
Spectrum | Flow Rate
(L min -1 ) | Detection
Limit
(Bq m -3 ) | Typical
uncertanty | Remote
Control | Need of dry
air sample | Need of corrections
depending on the
height of the inlet | Portability
Level and
monitor size | References                                                     |
|---------|------------------------------------------|---------------|-------------------------------------|---------------------------------------------|-----------------------|-------------------|---------------------------|----------------------------------------------------------------|------------------------------------------|----------------------------------------------------------------|
| ANSTO   | Dual-
flow-
loop
two-
filter | No            | ~83                                 | 0.03                                        | 8-12%                 | Yes               | No                        | No                                                             | Low ;
1.92 m 3             | Whittlestone and
Zahorowski (1998);
Brunke et al. (2002) |
| ARMON   | Electrost
atic
depositi
on      | Yes           | 1-2                                 | ~0.2                                        | 20%                   | Yes               | Yes                       | No                                                             | Medium;
0.18 m 3           | Grossi et al. (2012)                                           |
| HRM     | One-
filter                           | Yes           | 20                                  | ~0.05                                       | 15-20%                | Yes               | No                        | Yes                                                            | High;
0.08 m 3             | Levin et al. (2002)                                            |
| LSCE    | One-
filter                           | Yes           | 160                                 | ~0.01                                       | 20%                   | Yes               | No                        | Yes                                                            | High;
0.03 m 3             | Polian, 1986; Biraud,
2000                                  |

7. I suggest an additional figure with a synthesis of the slopes between the different monitors that are summarized in Table 2. This could be in the form of ANSTO vs. all of the other monitors for each site. However, keep table 2 as it contains all of the detail, it's just not easy to picture and visualize. I have added a figure to demonstrate what I mean.

Within the text of the point 7 the reviewer suggested to plot the results of Table 2 in the form 'ANSTO vs all' but in the figure he copied as example he plotted 'ARMON vs all'. We guess he meant the second case. The suggested figures, as summary of the table results, will be added to the manuscript both for ODM and SAC sites.

Technical comments Figures:

Sometimes hard to distinguish between the blue traces (ANSTO) and the black traces (ARMON) on the figures. However, this may be due to my eyes?

We have tried different colors. Here we copy an example with red (for us the best choice) used for the ARMON. We will change all figures in agreement with it within the modified version of our manuscript.

Line 42: replace "because of the" with "from the".

Lines 200 – 201: "method C". It's unclear what this means.

Line 251 and 252: I don't understand this sentence.

Line 251 - 257: Switched tense after the first sentence.

Line 255: Replace "Fine" with "fine"

Line 261: Replace "in order to" with "To"

Line 353 – 358: This long sentence is hard to follow. Please revise.

Line 383: Remove "compared" Use "" or "alpha" Use "progeny" or "daughters".

The previous changes suggested by the reviewer will be applied within the modified manuscript.

---

## Author Comment (AC3) · 20 Dec 2019

We want to thank Susana Barbosa for her review. The suggestions and comments have been considered. Answers (in blue color) to her specific comments are reported here. The recommended changes within the manuscript will be applied as soon as the open discussion will be ended.

Fig. 1: maybe add small arrows pointing to the inlets, particularly in case (c)

We added black arrows as suggested by the reviewer.

[Figure]

Section 2.3: the first sentence (lines 251-252) is not clear to me... I would also suggest specifying the height at which the meteorological measurements are taken, as well as the atmospheric aerosol concentration

The sentence has been changed to: 'Meteorological data used within this study were available from continuous measurements carried out at the SAC and ODM stations at 100 m and at 10 m a.g.l. respectively. The measurements were carried out with a Vaisala Weather Transmitter WXT520 (Campbell Scientific) for: (1) wind speed and direction (accuracies of ± 3 % and ± 3 ºC, respectively); (2) Humidity and temperature (accuracies of ± 3 % and ± 0.3 ºC, respectively). In addition, the atmospheric aerosol concentration was measured at ODM site using a fine dust measurement device Fidas® 200 S (Palas) at 10 m a.g.l.. The measurement range is between 0 and 20.000 particles cm$^{-3}$. All the accuracies refer to the manufacturer's specifications.'

Figure 2: possibly display also (maybe as supplemental material) the plot of the difference time series

The authors have discussed this suggestion and they think that may be will be more interesting plotting the time series of the ratios of $^{222}$Rn and $^{218}$Po measured by ANSTO, HRM and LSCE monitors again the $^{222}$Rn measured by the ARMON (i.e. these will represent the temporal change of the correction factors). The following figures will be added to the modified version of the support material.

[Figure]

Figure S1. Hourly time series of the ratios between the atmospheric $^{218}$Po and $^{222}$Rn activity concentration measured by each monitor (HRM, LSCE and ANSTO_ODM) and the one measured by the ARMON at Orme de Merisiers (ODM) station during Phase I (between 25 November 2016 and 23 January 2017).

[Figure]

Figure S2. Hourly time series of the ratios between the atmospheric $^{218}$Po and $^{222}$Rn activity concentration measured by each monitor (HRM and ANSTO_SAC) and the one measured by the ARMON at Saclay (SAC) station between 25 January 2017 and 13 February 2017.

Section 3.3: in my opinion it is not clear that data does not show any evident pattern...
for example, at least by eye, seems to me that LSCE and HRM values relative to ARMON as well as relative to ANSTO_ODM show a decreasing trend with temperature...

Actually a small influence has been observed at ODM as suggested by the reviewer. We will add the following paragraph to the modified version of the manuscript:

'Data does not show any evident patterns at 100 m a.g.l. (SAC station), which could indicate that there is any impact on $^{222}$Rn or $^{222}$Rn progeny measurements due to change of ambient temperature and relative humidity, at least not until saturated conditions are achieved. At 2 m a.g.l. (ODM station) a small decrease of about $10^{-2}$ $^{o}$C$^{-1}$ is observed in the ratio between the $^{214}$Po activity concentration (measured by HRM and LSCE monitors) and the $^{222}$Rn activity concentration (measured by ANSTO_ODM and ARMON monitors) when the increase of the ambient temperature. This temperature dependency may be rather due to the effect of atmospheric activity concentrations, increasing during nightime, on the disequilibrium between radon and its progeny. However, this influence on measured $^{214}$Po/$^{222}$Rn ratios seems to be quite small compared with others observed effects (ex.: loss of progeny within the sample tube, rain effects on radon progeny, atmospheric aerosol concentration).'

Page 16, line 421: maybe aerosol loading (instead of aerosol burden)

Change will be applied as suggested.

---

## Referee Comment (RC3) · Susana Barbosa (Referee) · 23 Dec 2019

I thank the authors for taking into account most of the issues raised in the original review. I would like to add the following points:

- concerning the suggestion of plotting the difference time series, I still would prefer to see a plot of time series differences. Actually in the comparison of time series the best practice is to show both absolute (differences) and relative (ratio) scales. In my opinion showing both absolute and relative results is justified and advisable in a inter-comparison study.

- given the relevance of the ARMON direct monitor in this inter-comparison study, its uncertainty should be clearly indicated. It is reported as 20% in Table 1, but in Figure

2 the measurements from the ARMON detector show large spikes which seem to be large than 2 Bq/m3...

---

## Author Comment (AC4) · 15 Jan 2020

Thanks the reviewer again for her time. We report here in attachment the reviewer comments and our specific answers (in blue color). The recommended changes within the manuscript will be applied as soon as the open discussion will be ended.

- concerning the suggestion of plotting the difference time series, I still would prefer to see a plot of time series differences. Actually in the comparison of time series the best practice is to show both absolute (differences) and relative (ratio) scales. In my opinion showing both absolute and relative results is justified and advisable in a inter-comparison study.

We agree with the reviewer that the best option is presenting both absolute (differences) and relative (ratio) time series. We will add the following Figures in the support material:

[Figure]

Figure S1. Hourly time series of the differences (a) and the ratios (b) between the atmospheric 222Rn or 218Po activity concentration measured by each monitor (HRM, LSCE and ANSTO_ODM) and the 222Rn measured by the ARMON at Orme de Merisiers (ODM) station during Phase I (between 25 November 2016 and 23 January 2017).

[Figure]

Figure S2. Hourly time series of the differences (a) and the ratios (b) between the atmospheric 222Rn or 218Po activity concentration measured by each monitor (HRM and ANSTO_SAC) and the 222Rn measured by the ARMON at Saclay (SAC) station between 25 January 2017 and 13 February 2017.

- given the relevance of the ARMON direct monitor in this inter-comparison study, its uncertainty should be clearly indicated. It is reported as 20% in Table 1, but in Figure 2 the measurements from the ARMON detector show large spikes which seem to be large than 2 Bq/m3...

The total uncertainty of the atmospheric radon concentration measured by the ARMON has been estimated to be of about 20% (k=2). This total uncertainty takes into account the uncertainty of the ARMON calibration factor $F_{Cal}$, the uncertainty related with humidity correction factor and the uncertainty on the net counts per minutes of detected $^{218}$Po. This last one, as reported in Grossi et al., 2012 and Vargas et al., 2015, is depending from the $^{218}$Po total counts and the 32% of total counts of $^{212}$Po decaying in $^{212}$Bi.

The ARMON has been calibrated within the INTE's radon chamber for a concentration interval ranging between $10^2$ Bq m$^{-3}$ to $10^3$ Bq m$^{-3}$ and an absolute humidity interval between $2 \cdot 10^2$-$2 \cdot 10^3$ ppm. The calibration factor $F_{Cal}$ has an estimated uncertainty of about 10% (k=2). The ARMON calibration, as well as the calibration of the other monitors participating in the inter-comparison campaign, was linearly extrapolated for lower atmospheric radon concentration values because

of the lack, so far, of a really low radon source and a robust traceability chain for low atmospheric radon concentration measurements.

The differences observed in Figure 2 and 3 of the manuscript could be due to a larger ARMON uncertainty for low atmospheric radon concentration measurements or to a smoothing effect of the ANSTO detector, due to its big volume, when fast changes occur in the atmospheric radon concentration. This should be better investigated in the next future thanks to long-term comparison campaigns and details analysis of the total monitors response uncertainties for low activity concentrations.

We have added within the manuscript three paragraphs in the methods, results and conclusion sections respectively:

'The calibration factor $F_{cal}$ of the ARMON used in this study was of 0.39 counts per minute (cpm) per Bq $m^{-3}$ with an uncertainty of about 10% (k=2). The total uncertainty of the atmospheric radon concentration activity measured by the ARMON takes also into account: the correction factor for the humidity influence inside the sphere was of $6.5 \cdot 10^{-5}$ per part per million $H_2O$ (ppm) and the uncertainty of the net α counts of $^{218}Po$'.

'Figure 2 and 3 show a larger hourly variability of the HRM and ARMON signals compared with the ANSTO ones. This difference in variability is attributable to the combination of a larger counting uncertainty of the HRM and ARMON detectors, and that only an approximated response time correction could be applied to the output of the ANSTO detectors (Griffiths et al. 2016). Further investigations should be carried out to clarify these differences and to exactly quantify the detectors uncertainties for the low $^{222}Rn$ concentrations typical for outdoor environmental monitoring at or above 100 m a.g.l. During the period of Jan 30 – February 1, 2019, the HRM shows significantly lower values than the ANSTO and ARMON. This period coincides with saturated air humidity conditions.'

'Finally, the direct ARMON seems to have a great potential for being used within atmospheric radon networks. In order to deeply evaluate the qualities and faults of this new instrument a long term inter-comparison study should be carried out using a direct ANSTO instrument.'.

---

## Author Response (AR1)

We would like to thank the two reviewers of this paper. All of their suggestions and comments have been considered for the improvement of the revised manuscript. Below, answers (in blue) to their specific comments are provided. Where relevant, here we also show the changes applied to the revised manuscript to comply with the recommendations.
* * *
Reviewer 1

Specific comments

1. I feel that a solid aspect of this paper is that the ARMON monitor performs extremely well and has excellent potential for deployment in radon networks. The other instruments have all been components of previous inter-comparison studies. Therefore, I suggest that the manuscript should be ARMON centric rather than being an inter-comparison study. I think that there is huge value in the work presented herein and the ARMON should be showcased. Perhaps change the title of the manuscript to reflect this?

We agree with the reviewer that the introduction of a new direct radon monitor, such as the ARMON, in the inter-comparison of radon/radon progeny monitors for atmospheric activity concentration measurements is the most solid aspect of this work. We also agree that this monitor seems to have a great potential to be used within radon networks. The measurement technique of the ARMON is not new because it was already applied in previous instruments such as one built at the Brazilian National Institute for Space Research (INPE) (Pereira and da Silva, 1989; Tositti et al., 2002). In addition, the ARMON monitors have been already used in the past years for different studies in the atmospheric research field (Grossi et al., 2012; Vargas et al., 2015; Hernandez-Ceballo et al., 2015; Grossi et al., 2016; Grossi et al., 2018).

However we would like to point out, from a general point of view, that this is the first time that four direct/indirect radon monitors, based on different measurement methods, have been compared in parallel at two measurement heights. This gives the opportunity of comparing their responses under the same atmospheric and meteorological conditions. It is also the first time that the performance of the ARMON has been compared with another direct radon monitor such as the ANSTO detector, which has been quite well characterized.

As correctly stated by the reviewer, the ARMON has a higher detection limit than the ANSTO detector, and a larger uncertainty. At the same time the ANSTO detector seems to slightly smooth the time series when fast changes in the atmospheric radon concentration are occurring. In order to correctly evaluate all these previous observations, the authors think it is necessary (and they are already planning), a long term inter-comparison campaign to specifically compare the performance of the ARMON and ANSTO detectors in detail, as explained in the Conclusions of this paper.

Therefore, in the present manuscript we would prefer to present the results of these comparisons between different monitors without focusing on any one instrument in particular.

However, the revised manuscript will showcase the introduction of another portable direct radon monitor, the ARMON, its potential, and the importance of completely evaluating its qualities and faults as a direct radon monitor for atmospheric stations as reported in the following new paragraphs:

Lines 391-397 of the revised manuscript: 'Figure 2 and 3 show a larger hourly variability of the HRM and ARMON signals compared with the ANSTO ones. This difference in variability is likely attributable a combination of a larger counting uncertainty of the HRM and ARMON detectors, and that only an approximated response time correction could be applied to the output of the ANSTO detectors (Griffiths et al. 2016) for the setup of this intercomparison. Further investigations should be carried out to clarify these differences and to exactly quantify the detector uncertainties for low $^{222}$Rn concentrations typical of outdoor environmental monitoring at or above 100 m a.g.l.'

Lines 534-536 of the revised manuscript: 'Finally, the direct new portable ARMON seems to have a great potential for being used within atmospheric radon networks. In order to deeply evaluate the qualities and faults of this new instrument a long term inter-comparison study should be carried out using a direct ANSTO instrument.'

2. In the abstract, the author mentions that this paper evaluates "correction factors between monitors". I think that the author needs to highlight that the slopes from the scatter plots are the correction factors.

This is correct and we will explicitly mention this in the revised manuscript as suggested by the reviewer:

Lines 33-34 of the revised manuscript: '…..linear regression fits between the monitors exhibited slopes, representing the correction factors,…'

3. I would like to see a section which compares the outcomes of this study with those from previous instrument comparisons (e.g. Schmithüsen et. al., 2017) to put the findings into context. How well do they agree? How site-specific are these corrections and what can be done to overcome this? What needs to be considered in future inter-comparison studies?

We agree that a section where the findings of this study are compared with those found in previous studies could be of interest. We decided here to compare the slopes/offsets of the regression lines calculated in this study between ANSTO and LSCE monitors against the HRM because they were also calculated in Schmithüsen et. al., 2017 for other ANSTO monitors and at different heights. The following section has been added to the revised manuscript:

**Lines 412-432 of the revised manuscript:**

**3.2 Comparison with past studies**

The results obtained in the present study of the slopes (b) and offsets (a) of the regression lines calculated between ANSTO or LSCE monitors against the HRM are here compared with the ones presented by Schmithüsen et. al., 2017. Table 3 shows a summary of this comparison. All slopes (correction factors) are defined as (routine station monitor) / HRM because this last was used as reference instrument by Schmithüsen et. al., 2017.

| Site/Input Height | Schmithüsen et al., 2017 | | | Present study | | |
|---|---|---|---|---|---|---|
| **ANSTO/HRM** | **Activity Range (Bq m$^{-3}$)** | **b** | **a** | **Activity Range (Bq m$^{-3}$)** | **b** | **a** |
| Cabauw: 200/180 m | 0-8 | 1.11±0.04 | 0.11±0.06 | | | |
| Saclay: 100 m | | | | 0-11 | 1.03±0.01 | 0.15±0.06 |
| Lutjewad: 60 m | 0-6 | 1.11 ± 0.02 | 0.11 ± 0.02 | | | |
| Heidelberg: 35 m | 0-15 | 1.22 ± 0.01 | 0.42 ± 0.04 | | | |
| Cabauw: 20 m | 0-12 | 1.30 ± 0.01 | 0.21 ± 0.03 | | | |
| Orme des Mérisiers: 2 m | | | | 0-22 | 1.17±0.01 | 0.63±0.03 |
| **LSCE/HRM** | **Activity Range (Bq m$^{-3}$)** | **b** | **a** | **Activity Range (Bq m$^{-3}$)** | **b** | **a** |
| Orme des Mérisiers: 2 m | 0-9 | 0.68±0.03 | -0.18±0.09 | 0-15 | 0.76±0.01 | -0.29±0.03 |

Data in Table 3 need to be analyzed taking into account that a unique traceability chain is not yet available for atmospheric radon measurements and the different monitors routinely running at the different stations could have different calibration chains (e.g. radon source, primary standard, etc.). Generally speaking, for both studies it can be observed that the correction factor between the atmospheric [214]Po activity concentration measured by HRM and the atmospheric [222]Rn activity concentration measured by ANSTO at each station approaches unity with the increase of the height of the sampling input. By contrast, the offsets of the regression fits decrease with the increase of the input height.

The only case where the compared instruments were exactly the same and at the same height is for Orme des Mérisiers station. Here the slope between the atmospheric [214]Po activity concentration measured by LSCE and HRM is equal to 0.76±0.01. This number is slightly larger but within uncertainties well comparable to the number reported by Schmithüsen et al. (2017) of 0.68±0.03 (see Table 3).

4. I think it would help to rearrange the methods section to clearly state that "direct" and "non-direct" methods are being compared. As highlighted above I feel that this is the really strong part of the manuscript as this brings in a second "direct" measurement.

We have rearranged the methods section as suggested by the reviewer.

5. Section 2.1.2. Can you add a little bit of information to describe how the measured progeny from the HRM one-filter monitor is related to 222Rn activity concentration? This is discussed in Schmithüsen et al (2017) but it would be good to see it repeated here.

We have added the following paragraph in the revised manuscript:

During the measurement campaign carried out at Saclay, where air samples were collected via a 100m Decabon tubing (see below), the line loss correction of Levin et al. (2017) was applied to all data of the HRM. No loss of aerosol was assumed in the short tubing used at Orme de Mérisiers station. Here we report for both sites [214]Po activity concentrations. However, for the 100 m intake height at Saclay we would not expect any disequilibrium, meaning that, based on the results from Schmithüsen et al. (2017), the reported [214]Po activity concentrations directly correspond to [222]Rn activity concentrations. By contrast, for the low 2 m intake at ODM we expect a [214]Po/[222]Rn disequilibrium of about 0.85 to 0.9.

6. Section 2.1.3. It is stated that the ARMON is portable. Can you elaborate and possibly give the dimensions?

We have added this information in the text and within the table 1.

Lines 208-209 of the revised manuscript: The detection volume of the ARMON is safety isolated because it is located within an external wood cube of 0.18 m$^3$.

| Monitor | Method | α Spectrum | Flow Rate (L min$^{-1}$) | Detection Limit (Bq m$^{-3}$) | Typical uncertainty (k=2) | Remote Control | Need of dry air sample | Need of corrections depending on the height of the inlet | Portability Level and monitor size | References |
|---|---|---|---|---|---|---|---|---|---|---|
| ANSTO | Dual-flow-loop two-filter | No | ~83 | 0.03 | 8-12% | Yes | No | No | Low ; 1.92 m$^3$ | Whittlestone and Zahorowski (1998) ; Brunke et al. (2002) |

| | | | | | | | | | | |
|---|---|---|---|---|---|---|---|---|---|---|
| **ARMON** | Electrostatic deposition | Yes | 1-2 | ~0.2 | 20% | Yes | Yes | No | Medium; 0.18 m³ | Grossi et al. (2012) |
| **HRM** | One-filter | Yes | 20 | ~0.05 | 15-20% | Yes | No | Yes | High; 0.08 m³ | Levin et al. (2002) |
| **LSCE** | One-filter | Yes | 160 | ~0.01 | 20% | Yes | No | Yes | High; 0.03 m³ | Polian, 1986; Biraud, 2000 |

7. I suggest an additional figure with a synthesis of the slopes between the different monitors that are summarized in Table 2. This could be in the form of ANSTO vs. all of the other monitors for each site. However, keep table 2 as it contains all of the detail, it's just not easy to picture and visualize. I have added a figure to demonstrate what I mean.

[Figure]

We have added Figure S7 in supplementary material to summarize the results of Table 2 for SAC and ODM stations using the ARMON as a reference.

[Figure]

Technical comments Figures:

Sometimes hard to distinguish between the blue traces (ANSTO) and the black traces (ARMON) on the figures. However, this may be due to my eyes?

We have tried a number of different colors to improve the readability of these graphs. Finally, we decided to use red for the ARMON data. All figures within the revised version of the manuscript have been changed in agreement with this.

Line 42: replace "because of the" with "from the".

Lines 200 – 201: "method C". It's unclear what this means.

Line 251 and 252: I don't understand this sentence.

Line 251 - 257: Switched tense after the first sentence.

Line 255: Replace "Fine" with "fine"

Line 261: Replace "in order to" with "To"

Line 353 – 358: This long sentence is hard to follow. Please revise.

Line 383: Remove "compared" Use " " or "alpha" Use "progeny" or "daughters".

The previous changes suggested by the reviewer have been applied in the revised manuscript.
* * *
Reviewer 2

Fig. 1: maybe add small arrows pointing to the inlets, particularly in case (c)

As suggested by the reviewer we added black arrows to figure 1 of the revised manuscript to indicate inlet positions.

[Figure]

Section 2.3: the first sentence (lines 251-252) is not clear to me... I would also suggest specifying the height at which the meteorological measurements are taken, as well as the atmospheric aerosol concentration

The paragraph has been modified in the revised manuscript:

Lines 286-293: Meteorological data used within this study were available from continuous measurements carried out at the SAC and ODM stations at 100 m and at 10 m a.g.l. respectively. The measurements were carried out with a Vaisala Weather Transmitter WXT520 ·(Campbell Scientific) for: (1) wind speed and direction (accuracies of $\pm$ 3 % and $\pm$ 3 ℃,

respectively); (2) Humidity and temperature (accuracies of ± 3 % and ± 0.3 ºC, respectively). In addition, the atmospheric aerosol concentration was measured at ODM site using a fine dust measurement device Fidas® 200 S (Palas) at 10 m a.g.l.. The measurement range is between 0 and $20·10^3$ particles $cm^{-3}$. All the accuracies refer to the manufacturer's specifications.'

Figure 2: possibly display also (maybe as supplemental material) the plot of the difference time series

The hourly time series of the differences and the ratios of $^{222}$Rn and $^{218}$Po measured by ANSTO, HRM and LSCE monitors again the $^{222}$Rn measured by the ARMON have been presented in Figures S1 and S2 of the supplemental material.

[Figure]

Figure S1. Hourly time series of the differences (a) and the ratios (b) between the atmospheric $^{222}$Rn or $^{218}$Po activity concentration measured by each monitor (HRM (green circles), LSCE (orange circles) and ANSTO_ODM (blue circles)) and the $^{222}$Rn measured by the ARMON at Orme de Merisiers (ODM) station during Phase I (between 25 November 2016 and 23 January 2017).

[Figure]

Figure S2. Hourly time series of the differences (a) and the ratios (b) between the atmospheric $^{222}$Rn or $^{218}$Po activity concentration measured by each monitor (HRM (green circles) and ANSTO_SAC (blue circles)) and the $^{222}$Rn measured by the ARMON at Saclay (SAC) station between 25 January 2017 and 13 February 2017.

Section 3.3: in my opinion it is not clear that data does not show any evident pattern...
for example, at least by eye, seems to me that LSCE and HRM values relative to ARMON as well as relative to ANSTO_ODM show a decreasing trend with temperature...

The reviewer was right. A small influence has been observed at ODM as suggested by the reviewer. The following paragraph has been added within the revised version of the manuscript and a Figure S8 has been presented within the supplemental material.

Lines 448-465: 'Data does not show any evident patterns at 100 m a.g.l. (SAC station), which could indicate that there is any impact on $^{222}$Rn or $^{222}$Rn progeny measurements due to change of ambient temperature and relative humidity, at least not until saturated conditions are achieved. By contrast, a small decrease, of about $10^{-2}$ °C$^{-1}$, is observed in the ratio between the $^{214}$Po activity concentration (measured by HRM and LSCE monitors) and the $^{222}$Rn activity concentration (measured by

ANSTO_ODM and ARMON monitors) with the increase of the ambient temperature (Figure S8 of the support material) at 2 m a.g.l. (ODM station). This temperature dependency may be attributable to the effect of atmospheric activity concentrations, increasing during nightime, on the disequilibrium between radon and its progeny. However, this influence on measured $^{214}$Po/$^{222}$Rn ratios seems quitesmall compared with other observed effects (e.g.: loss of progeny within the sample tube (Levin et al., (2017)), atmospheric aerosol concentration (see below)).'

Page 16, line 421: maybe aerosol loading (instead of aerosol burden)

Change has been applied in the revised manuscript.

- given the relevance of the ARMON direct monitor in this inter-comparison study, its uncertainty should be clearly indicated. It is reported as 20% in Table 1, but in Figure 2 the measurements from the ARMON detector show large spikes which seem to be large than 2 Bq/m3...

The total uncertainty of the atmospheric radon concentration measured by the ARMON has been estimated to be of about 20% (k=2). This total uncertainty takes into account the uncertainty of the ARMON calibration factor $F_{Cal}$, the uncertainty related with humidity correction factor and the uncertainty on the net counts per minutes of detected $^{218}$Po. This last one, as reported in Grossi et al., 2012 and Vargas et al., 2015, is depending from the $^{218}$Po total counts and the 32% of total counts of $^{212}$Po decaying in $^{212}$Bi.

The ARMON has been calibrated within the INTE's radon chamber for a concentration interval ranging between $10^2$ Bq m$^{-3}$ to $10^3$ Bq m$^{-3}$ and an absolute humidity interval between $2 \cdot 10^2$-$2 \cdot 10^3$ ppm. The calibration factor $F_{Cal}$ has an estimated uncertainty of about 10% (k=2). The ARMON calibration, as well as the calibration of the other monitors participating in the inter-comparison campaign, was linearly extrapolated for lower atmospheric radon concentration values because of the lack, so far, of a really low radon source and a robust traceability chain for low atmospheric radon concentration measurements.

The differences observed in Figure 2 and 3 of the manuscript could be due to a larger ARMON uncertainty for low atmospheric radon concentration measurements or to a smoothing effect of the ANSTO detector, due to its big volume, when fast changes occur in the atmospheric radon concentration. This should be better investigated in the near future thanks to long-term comparison campaigns and detailed analysis of the total monitors response uncertainties for low activity concentrations.

We have added within the revised version of our paper the following paragraphs:

Lines 203-209: 'The total uncertainty of the atmospheric radon activity concentration measured by the ARMON is of about 20% (k=2) where it is including the calibration factor $F_{cal}$, the background due to the presence of $^{212}$Po from $^{220}$Rn and the humidity correction factor (Grossi et al., 2012; Vargas et al., 2015).'

Lines 395-401: 'Figure 2 and 3 show a larger hourly variability of the HRM and ARMON signals compared with the ANSTO ones. This difference in variability is likely due to a larger uncertainty of the HRM and ARMON detectors and that only an approximated form of the Griffiths et al. (2016) response time correction was able to be applied to the ANSTO detectors in this study due to lack of information gathered during their setup. Further investigations should be carried out to clarify these differences and to

exactly quantify the detectors uncertainties for the low $^{222}$Rn concentrations typical for outdoor environmental monitoring at or above 100 m a.g.l.'

Lines 538-540: 'Finally, the new portable ARMON seems to have a great potential for being used within atmospheric radon networks. In order to deeply evaluate the qualities and faults of this new instrument a long term inter-comparison study should be carried out using a direct ANSTO instrument.'

---

## Editor Decision (ED1)

The authors have done a good job of responding to the reviewers' comments – thank you.

**Some further comments that should be looked at in a revised manuscript:**

1) Table 1 is potentially useful but could be improved. Some of the points here also relate to broader discussion in the text on uncertainty.

- "Sampling flow rate" instead of "flow rate"
- Detection limit for ANSTO is 0.03 but an approximation is given for the others. Why is this? A footnote explaining, e.g. differences in definitions, would be good.
- Over what range is the stated uncertainty relevant? This is important for matching the right instrument to the right application or measurement location. A lot of the measurements in the paper's time series border on the detection limit of the ARMON. Is a 20% uncertainty the case at 0.3 Bqm-3, for instance? For the ANSTO there is a discussion of some of this on page 5 "a counting uncertainty of around 2% for radon concentrations ≥1 Bq m-3", and a discussion for the HRM at the bottom of page 6. The discussion of uncertainties and what is stated in the table needs to be completely transparent for comparison between instruments.
- The portability column could be improved. A grading such as low/high might not be useful. Instead call this "portability considerations" and let the potential user decide based on their specific circumstances. Please state the three measured dimensions of each instrument in the description rather than a volume (which is difficult to physically relate to), and add the mass of the instruments – this is obviously very important too in terms of transportation and handling.
- Alongside portability is "deployability" i.e. level of automation, consumables required, energy consumption, which might be of even greater interest than portability. The basic monitor also needs peripherals e.g. large pumps, cryocoolers etc.

2) The conclusions and abstract need rephrasing and tightening up. Some things below but not exhaustive.
- The last sentence on page 17 is very confusing. What is "close to one" – the regression line? But that is not referred to in the sentence.
- "last behaviour" change to "the latter"
- Line 463 "very good" to "significant"
- Line 464 "slope of this correlation". This correlation discussion is confusing given the stated small uncertainties on the slopes stated alongside "within uncertainties well comparable". Please explain.
- "underlines that to assure".. "is important" – revise sentence structure.
- So does the ARMON help to meet the requirements on lines 476-480? It is stated that the ARMON has great potential but not why specifically in relation to what is needed in networks. Can you explain why further inter-comparison with the ANSTO is needed?
- Line 34 "daily basis". Not sure what this means – daily averages or within days?
- Lines 42 to 44 refer to the same points made at the end of the conclusion. This leaves the reader unclear as to what has been advanced in this work and what is needed next.

Specifics:

"close to and further up" change to "when sampling at 2 and 100 magl"

**Minor corrections/explanations needed:**

Page 5 line 164 "measurement uncertainty"

Page 5 mentions "detection limit", page 6 mentions "minimum detectable activity". This should be consistent throughout if these are referring to the same thing.

Table 1 "Need of .. height of inlet" could just be "Sampling inlet height correction"

Table 1 Uncertainty of HRM is 15-20% but in text <20%. Just be consistent with these reported values throughout the text so that the instruments can really be compared.

Page 6 line 190 – give details of the cryocooler

Page 17 – make space between number and unit.. 100 m.. 2 m etc

Page 13 – what is the approximated response time correction?

---

## Author Response (AR2)

Some further comments that should be looked at in a revised manuscript:

1) Table 1 is potentially useful but could be improved. Some of the points here also relate to broader discussion in the text on uncertainty.

- "Sampling flow rate" instead of "flow rate"

It has been changed

- Detection limit for ANSTO is 0.03 but an approximation is given for the others. Why is this? A footnote explaining, e.g. differences in definitions, would be good.

The ARMON's detection limit was calculated following Gilmore, 2008, as reported in Grossi et al., 2012 with a confidence level of 95%. The Detection Limits of the others instruments were calculated as presented in their reference papers (reported in the last column of the table for more details) as the ambient radon concentration at which the estimated counting error reaches 10% (Levin et al., 2002) and 30% (Chambers et al., 2016). We have now used this latter to harmonize the table column of the detection limits for all instruments.

It is important to underline that in the present study we did not measure the radon concentration background of each instrument for harmonizing the calculation of the detection limits properly because the inter-comparison campaign was carried out in field conditions and it was out of the scope of the study.

- Over what range is the stated uncertainty relevant? This is important for matching the right instrument to the right application or measurement location. A lot of the measurements in the paper's time series border on the detection limit of the ARMON. Is a 20% uncertainty the case at 0.3 Bqm-3, for instance? For the ANSTO there is a discussion of some of this on page 5 "a counting uncertainty of around 2% for radon concentrations ≥1 Bq m-3", and a discussion for the HRM at the bottom of page 6. The discussion of uncertainties and what is stated in the table needs to be completely transparent for comparison between instruments.

The total expanded uncertainties of all monitors have been now presented coherently with k = 2. The ARMON was calibrated within the INTE-UPC chamber at a range of hundreds Bq m$^{-3}$ because, so far, European radon chamber facilities are not able to create low level radon reference air. An uncertainty of <35% (k=2) was estimated for atmospheric radon levels of few Bq m$^{-3}$. A sentence has been included to explain this. In addition in the next future we want to carry out a long term intercomparison campaign, in the mark of the new EMPIR project traceRadon, in order to correctly harmonize and calculate the uncertainties of HRM, ANSTO and ARMON monitors using the same reference radon air and background.

- The portability column could be improved. A grading such as low/high might not be useful. Instead call this "portability considerations" and let the potential user decide based on their specific circumstances. Please state the three measured dimensions of each instrument in the description rather than a volume (which is difficult to physically relate to), and add the mass of the instruments – this is obviously very important too in terms of transportation and handling.

As suggested by the reviewer we have reported these values within the column and changed the column names. However each instrument is composed from different parts, not only the main detection volume. There are also pump, filters or electronics components. Depending on the instrument. All such details are already reported in the reference papers of each instrument.

- Alongside portability is "deployability" i.e. level of automation, consumables required, energy consumption, which might be of even greater interest than portability. The basic monitor also needs peripherals e.g. large pumps, cryocoolers etc.

A new column has been created in the table where we have reported the main needs of each instrument (dry air, possibility to check the spectrum, remote connection, etc.). Other aspects such as filter, maintenance, etc. are interesting and they have been reported in more detail in the text because within the table columns there is not enough space. None of the instruments consume a large amount of energy, so this does not make a significant difference between them.

2) The conclusions and abstract need rephrasing and tightening up. Some things below but not exhaustive.

We have worked on improving the conclusions and the abstract. We have also applied the suggested changes.

- The last sentence on page 17 is very confusing. What is "close to one" – the regression line? But that is not referred to in the sentence.

We have changed the sentence.

- "last behaviour" change to "the latter"

The change has been applied

- Line 463 "very good" to "significant"

The change has been applied

- Line 464 "slope of this correlation". This correlation discussion is confusing given the stated small uncertainties on the slopes stated alongside "within uncertainties well comparable". Please explain.

Here we were comparing the slope of the regression fit calculated between the LSCE and the HRM monitors at ODM station during this study with the same slope calculate by Schmithüsen et al. (2017). We have changed the sentence.

- "underlines that to assure".. "is important" – revise sentence structure.

The sentence has been changed.

- So does the ARMON help to meet the requirements on lines 476-480? It is stated that the ARMON has great potential but not why specifically in relation to what is needed in networks. Can you explain why further inter-comparison with the ANSTO is needed?

The text has been improved in order to clarify these points.

- Line 34 "daily basis". Not sure what this means – daily averages or within days?

We mean that the monitors were all able to observe the changes of the atmospheric radon concentration during the day: the nocturnal accumulation of the radon concentration during the night due to the shallow planetary boundary layer and the diurnal dilution of the concentration due to increase of the turbulence.

- Lines 42 to 44 refer to the same points made at the end of the conclusion. This leaves the reader unclear as to what has been advanced in this work and what is needed next.

The sentence has been changed to better differentiate between what has been done here and what still needs to be done.

Specifics:
"close to and further up" change to "when sampling at 2 and 100 magl"

It has been changed

**Minor corrections/explanations needed:**
Page 5 line 164 "measurement uncertainty"
Corrected

Page 5 mentions "detection limit", page 6 mentions "minimum detectable activity". This should be consistent throughout if these are referring to the same thing.

This has been clarified in the beginning of this document and it has been corrected in the text.

Table 1 "Need of .. height of inlet" could just be "Sampling inlet height correction"

It has been changed accordingly

Table 1 Uncertainty of HRM is 15-20% but in text <20%. Just be consistent with these reported values throughout the text so that the instruments can really be compared.

Values have been reported coherently in Table 1 now.

Page 6 line 190 – give details of the cryocooler

A sentence on this has been added within the text with more reference to past studies.

Page 17 – make space between number and unit.. 100 m.. 2 m etc

It has been done

Page 13 – what is the approximated response time correction?

For the deconvolution routine of Griffiths et al (2016) to be run in its intended form it is necessary to perform a source "spike test" at the sampling inlet (so that the combined characteristics of the whole intake system, delay volume and detector can be taken into account by the model). Unfortunately, we were not able to perform a "spike test" on the detectors installed for this inter-comparison campaign, so we estimated the characteristics based on what we knew of the setup, and performance of similar detectors. A small paragraph has been added to clarify this point.

[revised manuscript text omitted]

 • Remote control
 • Sampling inlet height correction | Levin et al. (2002) |
| LSCE | One-filter | ~160 | ~0.01 | < 20% | High; 0.03 m² 25x25x40 ~8 | • α Spectrum
 • Remote control
 • Sampling inlet height correction
 • Need of large pump | Polian, (1986); Biraud, (2000) |

Table 1. Summary of principal characteristics of the $^{222}$Rn and $^{222}$Rn progeny monitors compared in the present study.

**2.2 Sites**

The present inter-comparison study was carried out at two stations located 30 km southwest of Paris in the fall and winter of 2016-2017 (Figure 1). Both stations, 3.5 km apart, belong to the LSCE and are located in a region with a radon flux of ca. 5-10 mBq m$^{-2}$ s$^{-1}$ in winter, according to output of the Karsten et al. (2015) model.

Phase I of the measurements started at Orme des Mérisiers (ODM, latitude 48.698, longitude 2.146, 167 m above sea level) and ran between 25 November 2016 and 23 January 2017. Here, LSCE and ANSTO (for convenience named here as ANSTO_ODM) monitors are routinely running. During Phase I of the

**Commented [CS1]:** Performance characteristics of the 1500L detectors have changed considerably in the past 20 years – so I suggest including at least one of the more recent papers that discuss their performance in more detail. Examples include Chambers et al (2011, 2014, 2018). I think just the 2018 paper would be enough, since it mentions the others.

[revised manuscript text omitted]

of the ANSTO detectors. This are taken into account the day in the ANSTO response into big detection volume. 
[revised manuscript text omitted]